# Similarities and differences in patterns of germline mutation between mice and humans

Sarah J. Lindsay [1], Raheleh Rahbari [1], Joanna Kaplanis[1], Thomas Keane [1] & Matthew E. Hurles[1]

Whole genome sequencing (WGS) studies have estimated the human germline mutation rate per basepair per generation (~$1.2 \times 10^{-8}$) to be higher than in mice ($3.5$–$5.4 \times 10^{-9}$). In humans, most germline mutations are paternal in origin and numbers of mutations per offspring increase with paternal and maternal age. Here we estimate germline mutation rates and spectra in six multi-sibling mouse pedigrees and compare to three multi-sibling human pedigrees. In both species we observe a paternal mutation bias, a parental age effect, and a highly mutagenic first cell division contributing to the embryo. We also observe differences between species in mutation spectra, in mutation rates per cell division, and in the parental bias of mutations in early embryogenesis. These differences between species likely result from both species-specific differences in cellular genealogies of the germline, as well as biological differences within the same stage of embryogenesis or gametogenesis.

[1] Wellcome Sanger Institute, Hinxton, Cambridge CB10 1SA, UK. Correspondence and requests for materials should be addressed to M.E.H. (email: meh@sanger.ac.uk)

Most germline single-nucleotide mutations in humans (75–80%) are paternal in origin, and increasing paternal age is the major factor determining variation in numbers of mutations per offspring in humans[1–3] with an average increase of 1–2 paternal de novo mutations (DNMs) per year. Recently, a more modest effect of maternal age has been reported, equating to an additional 0.24–0.5 DNMs per year[4,5]. However, parental age effects, and other factors that influence variation in germline mutation rate, have not been well characterised in other species. The paternal age effect has been attributed to the high number of ongoing cell divisions, and concomitant genome replications, in the male germline[2,3]. However, as the ratio of the number of paternal and maternal germline cell divisions in humans considerably exceeds the ratio of paternal and maternal-derived mutations[6], it appears not all germline cell divisions are equally mutable.

Germline mutations can arise at any stage of the cellular lineage from zygote to gamete. Spermatogonial stem cell (SSC) divisions in post-pubertal males account for the highest proportion of all cell divisions in the germline. Mutations that arise in the first ~10 cell divisions prior to the specification of primordial germ cells (PGCs) can be shared with somatic lineages. In humans, at least 4% of de novo germline mutations are mosaic in parental somatic tissues[3,7]. Mutations that arise just after PGC specification should lead to germline-specific mosaicism, although the typically small numbers of human offspring per family limit the detection of germline mosaicism and thus our understanding of mutation processes post-PGC specification. Studies of phenotypic markers in mice have suggested variability in mutation rates and spectra at different stages in the germline[8–10], and mutational variability between germline stages has been implicated in recent work in humans[3], cattle[11] and drosophila.[12]

To characterise mutation rates, timing and spectra in the murine germline, we analyse the patterns of DNM sharing among offspring and parental tissues in six large mouse pedigrees (Fig. 1), using a combination of whole-genome sequencing (WGS), deep targeted sequencing and an analytical workflow described previously[3]. We then compare these murine patterns with equivalent, previously published, human data on three multi-sibling families[3]. We find that mutation rates in both species are highest in the first cell division that contributes to the embryo. We note a parental age effect in mice, also driven largely by mutations in the male germline. We observe higher mutation rates in male mice compared to female mice during early embryogenesis, whereas in humans, the mutation rate is comparable between the sexes. We observe differences in the spectra of DNMs in mice and humans, driven by an increase in $T \rightarrow A$ and a decrease in $T \rightarrow C$ mutations in mice. Finally, we find that mutation rates in SSC cell divisions are lower in humans than in mice.

## Results

**Detection of DNMs in mice.** After WGS, discovery and validation of candidate DNMs in 5 or 10 offspring from each mouse pedigree, we validated 753 unique autosomal DNMs (746 single-nucleotide variants (SNVs) and 7 multinucleotide variants (MNVs), with a range of 8–36 de novo SNVs per offspring (Supplementary Table 1). We sequenced all validated DNMs in three tissues from the parents (mean coverage of 400–800× per tissue), two tissues from the WGS offspring (mean coverage of 400×) and a single tissue from all other offspring (mean coverage of 200×) (Fig. 1). We determined that 2.7-fold more unique DNMs were of paternal ($N = 152$) than of maternal ($N = 55$) origin, similar to previous studies[13,14]. Mice and humans have more similar paternal biases in mutations than might be expected

(2.7:1 and 3.9:1, respectively[1,3–5]), given the fivefold difference in the ratios of genome replications in the paternal and maternal germlines between mice (2.5:1) and humans (13:1)[6]. We did not observe any significant strain-specific differences between the reciprocal crosses and so combined data from these crosses in downstream analyses.

In the two largest mouse pedigrees, we observed that 18% (70/388) of unique DNMs were shared among 2–19 siblings, strongly implying a single ancestral mutation. This observation suggests that an appreciable proportion of DNMs in mice derive from early mutations (and therefore germline mosaicism) in the parental germline.

**Mutation timing.** For both species, using the patterns of mutation sharing across the parental and offspring tissues, we classified DNMs into four temporal strata of the germline (Fig. 2a, b).

We refer to these four strata as very early embryonic (VEE), early embryonic (EE), peri-primordial germ cell specification (peri-PGC) and late post-primordial germ cell specification (late post-PGC). VEE mutations likely arose within the earliest post-zygotic cell divisions contributing to the developing embryo and are characterised by a low variant allele fraction (VAF) in the offspring and are absent from parental tissues, with consistent representation in 25–50% of cells across two offspring tissues (Supplementary Fig. 1), reflecting their likely origin within the earliest post-zygotic cell divisions contributing to the developing embryo. As VEE mutations are detected in the offspring and arise in the embryo before the lineages for the germline and soma are specified, in principle VEE mutations could be restricted to the germline or soma or could be shared by both. EE mutations were defined as apparent DNMs observed constitutively in offspring and mosaic in parental somatic tissues, typically mosaic in a lower proportion of cells (2–20%) than VEE mutations, consistent with them arising during later embryonic cell divisions, prior to PGC specification (after ~10 cell divisions). Peri-PGC mutations were observed shared among two or more offspring of the same parents and so are likely parental germline mosaic but are not detectably mosaic in parental somatic tissues (<1.6% of cells), compatible with them arising around the time of PGC specification and the separation of germline and somatic lineages. Unlike in humans, mouse PGC specification is well characterised; after specification, PGCs proliferate rapidly, generating thousands of germ cell progenitors in both sexes[15–18]. In the absence of strong positive selection, only mutations that occur prior to this proliferation are likely to be observed in multiple siblings in our pedigrees. In support of this assumption, studies of phenotypic markers of mutation have indicated that SSCs need to be depleted almost to extinction to result in sharing of phenotypes induced by later mutations among offspring. Finally, late post-PGC mutations are observed constitutively in a single offspring and represent mutations arising during cell divisions from PGC proliferation and gametogenesis. Collectively, these four temporal strata represent a full generational span (Fig. 2b), albeit with the VEE mutations being quantified here in offspring. VEE mutations accounted for 23.9% (194/811) of all observed DNMs in mice but only 4% in humans (28/719) (Fig. 3). VAFs for the observed VEE mutations in mice and humans were consistent with the vast majority occurring in the first cell division that contributes to the embryo and were highly concordant between different tissues. The number of VEE mutations per offspring varied considerably more than expected under a Poisson distribution ($p = 0.0019$), suggesting that this stage is more mutagenic for some zygotes than others. This latter observation implies that most VEE mutations are not due to later somatic mutation. (Supplementary Table 1, Supplementary Fig. 1).

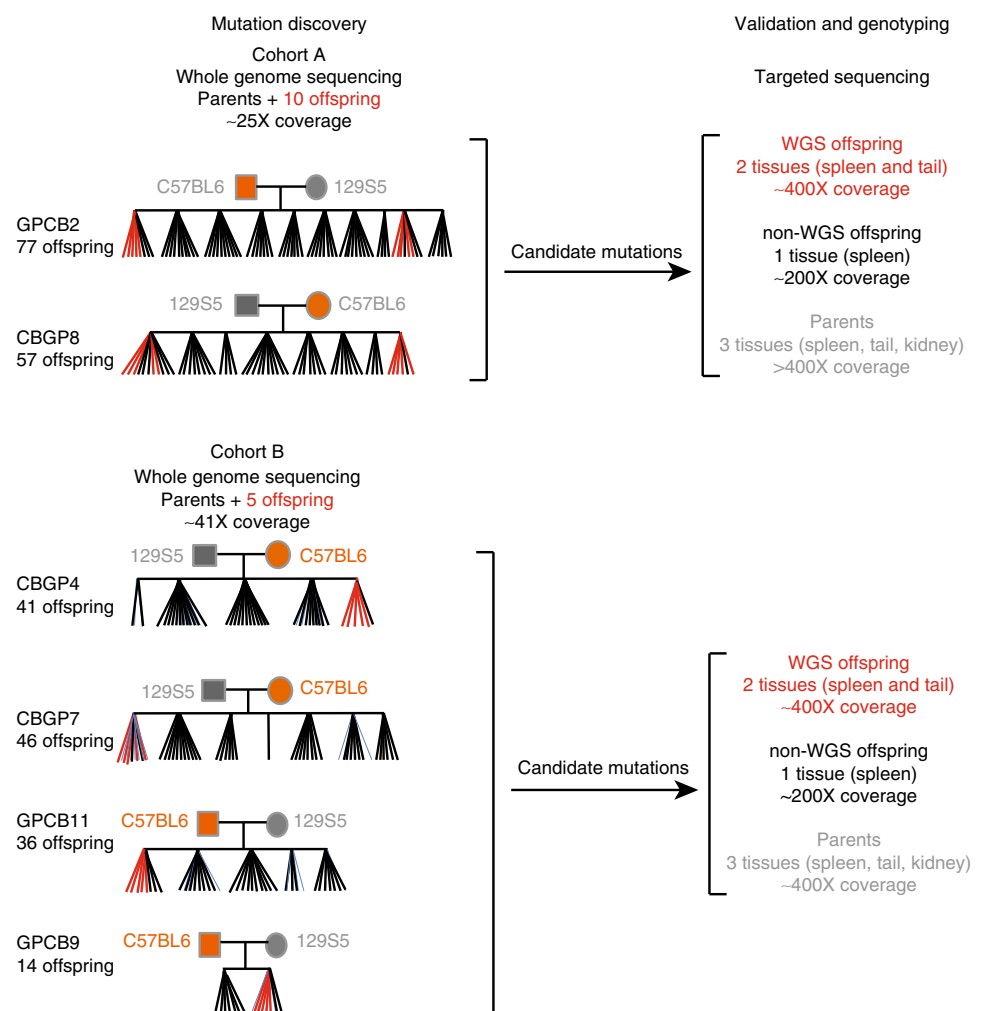

**Fig. 1** Mouse pedigree sequencing and genotyping strategy. Six reciprocal crosses were established and successively mated, generating six pedigrees comprising between 41 and 77 offspring. The pedigrees shown here only include the sequenced and genotyped individuals. Three tissues (spleen, kidney and tail) were collected from all mice. Whole-genome sequencing (WGS) was performed on spleen-derived DNA for five or ten pups within each pedigree (shown in red). Candidate DNMs were identified and validated using deeper targeted sequencing of 1–3 tissues per individual across the offspring shown. The naming conventions and differences between data collection and analysis are shown. Not all offspring from each pedigree were genotyped

The 55 EE mutations we detected in mice were present at similar levels across three parental somatic tissues (1.6–19%), and we observed a significant but modest correlation between levels of somatic and germline mosaicism (Pearson correlation coefficient of 0.40, $p = 0.0025$, Supplementary Figs. 1 and 2, Supplementary Table 2). The proportion of DNMs observed as EE mutations in mice represented a very similar proportion of all of EE mutations observed in human pedigrees[3]. We observed a significant paternal bias among the EE mutations identified in mice (41 paternal, 14 maternal, $p = 0.0004$, two-sided binomial test) but not in humans (9 paternal, 16 maternal, $p = 0.23$, two-sided binomial test), and this difference between species is significant ($p = 0.002$, Fisher's Exact Test). The correlation between germline and somatic mosaicism was notably stronger among paternal EE mutations than among maternal EE mutations and is suggestive of sex differences in the cellular genealogy of EE development (Supplementary Table 2). We tested the dependence of the EE sex difference on families and found that modelling including an individual effect neither improves a model of parent–EE mutations nor is a significant predictor on its own ($p = 0.15$, mixed effects regression).

We considered and discounted a wide variety of possible technical artefacts that might explain this apparent parental sex

bias in EE mutations in mice ('Methods'). We propose two possible biological explanations for this paternal bias in mice: (i) an elevated paternal mutation rate per cell division or (ii) a later paternal split between soma and germline (i.e. more shared cell divisions). Further work is required to define these potential sex-specific differences in germline genealogy, although the observation of early sex dimorphism in pre-implantation murine and bovine embryos[11,19] may well be relevant. The high proportion (average 30%) of DNMs in mice that appear to occur during EE development (VEE+EE mutations) is in accordance with high estimates of germline mosaicism from phenotypic studies[8–10].

We identified 54 peri-PGC DNMs (shared among siblings but not detectable in parental somatic tissues) in the two largest mouse pedigrees but with no sex bias in parental origin, where it could be determined (25 maternal, 25 paternal), suggesting that the sex differences identified for EE mutations are confined to a small developmental window. We did not observe any preferential sharing of these DNMs within litters as opposed to between litters (Fig. 3), as might be expected if only a subset of SSCs were productive at any one time. Unlike EE mutations, peri-PGC mutations arose approximately equally in the paternal and maternal germlines (direct phasing: 6 paternal, 8 maternal; inferred parental origin using co-occurrence: 25 paternal, 25

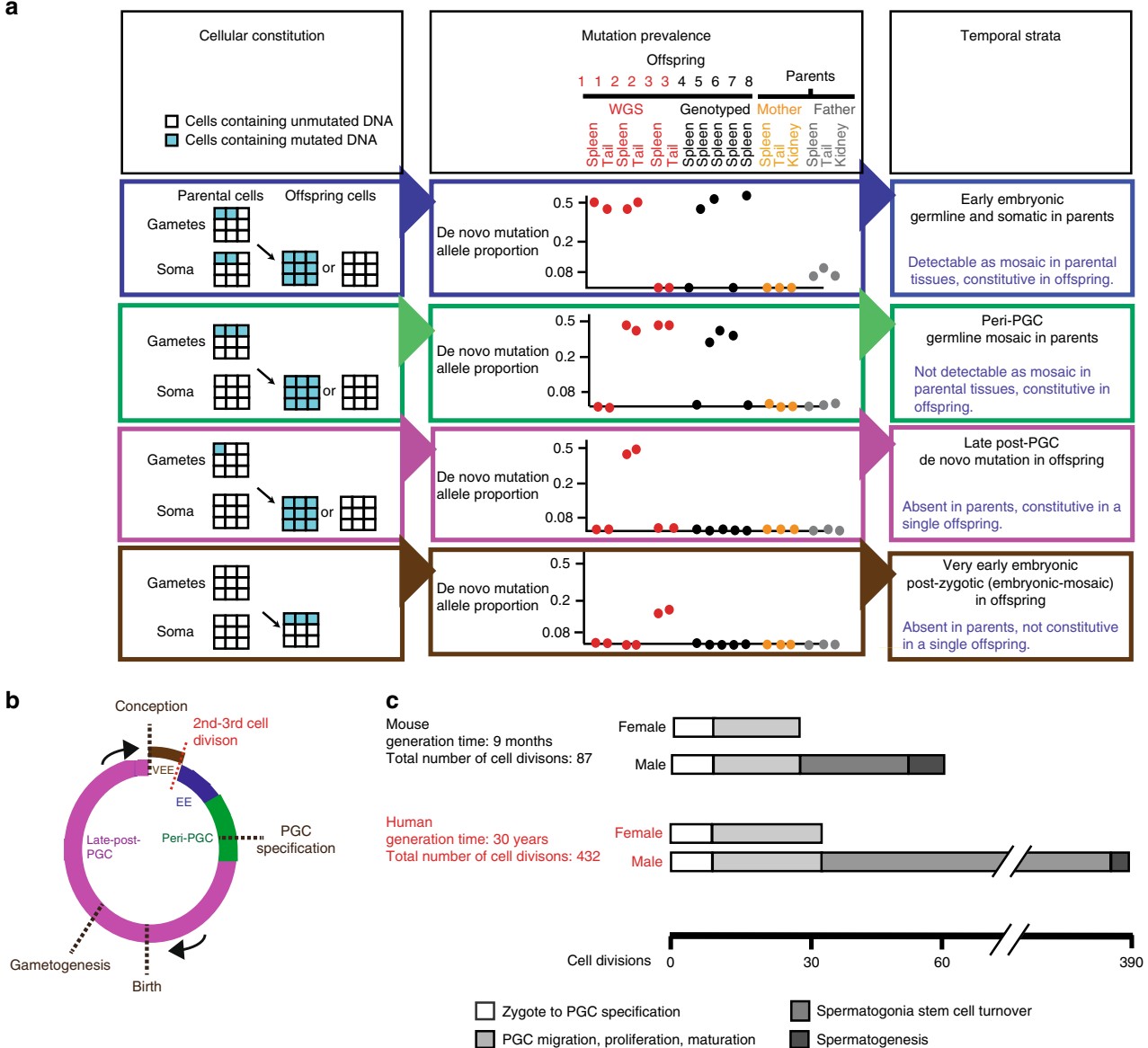

**Fig. 2** Temporal strata of observed mutations. **a** (Left) Four classes of mutations distinguished by the proportion of cells carrying the mutation within parental and offspring tissues. Multiple tissues were used to verify consistency of VAF. (Middle) Illustrative examples of variant allele fraction observed among the sequenced samples for each class. (Right) Temporal strata to which these mutations are assigned. **b** Four temporal strata mapped onto the generational cycle of the germline: EE, peri-PGC and late post-PGC mutations were detected in parents, VEE mutations were detected in offspring. **c** Cell divisions occurring in the different stages of the mice and human germlines during an average generation[6]

maternal). We detected 31 peri-PGC DNMs in the four smaller pedigrees and only observed 4 peri-PGC DNMs in the human pedigrees. The numbers are not directly comparable between species and pedigrees, due to the disparity in numbers of offspring per pedigree and therefore the power to distinguish this class of DNMs.

**Comparing mouse and human germline mutation rates.** For both species, we estimated germline autosomal mutation rates per generation, per year and per cell division (Table 1), accounting for the partial contribution of mosaic (VEE) mutations to the germline ('Methods'). To estimate the contribution of VEE mutations detected in somatic tissues to the germline, we identified and validated 26 candidate VEE mutations in parental genomes. Twenty-one of these validated VEE mutations were transmitted to offspring. We observed a significant correlation between the VAF of the VEE mutation and the proportion of

offspring to which the mutation was transmitted (Pearson correlation = 0.71, $p = 0.00011$; 'Methods', Supplementary Fig. 3, Supplementary Table 3). We fitted a linear model to the relationship between somatic and germline mosaicism, which enabled us to estimate the germline contribution of each VEE mutation. On average, this calculation reduced the contribution of VEE mutations to the germline mutation rate by 40%.

After inferring the contribution of VEEs to the germline rate and accounting for our sensitivity to detect DNMs in all six pedigrees ('Methods'), we estimated the mutation rate per generation in mice to be approximately one third to that in humans, whereas the annual mutation rate was 13 times higher and the per cell division mutation rate was more than one and a half times higher.

The 13-fold difference in annual mutation rates between extant mouse and human is substantially greater than the approximately 2-fold greater accumulation of mutations on the mouse lineage

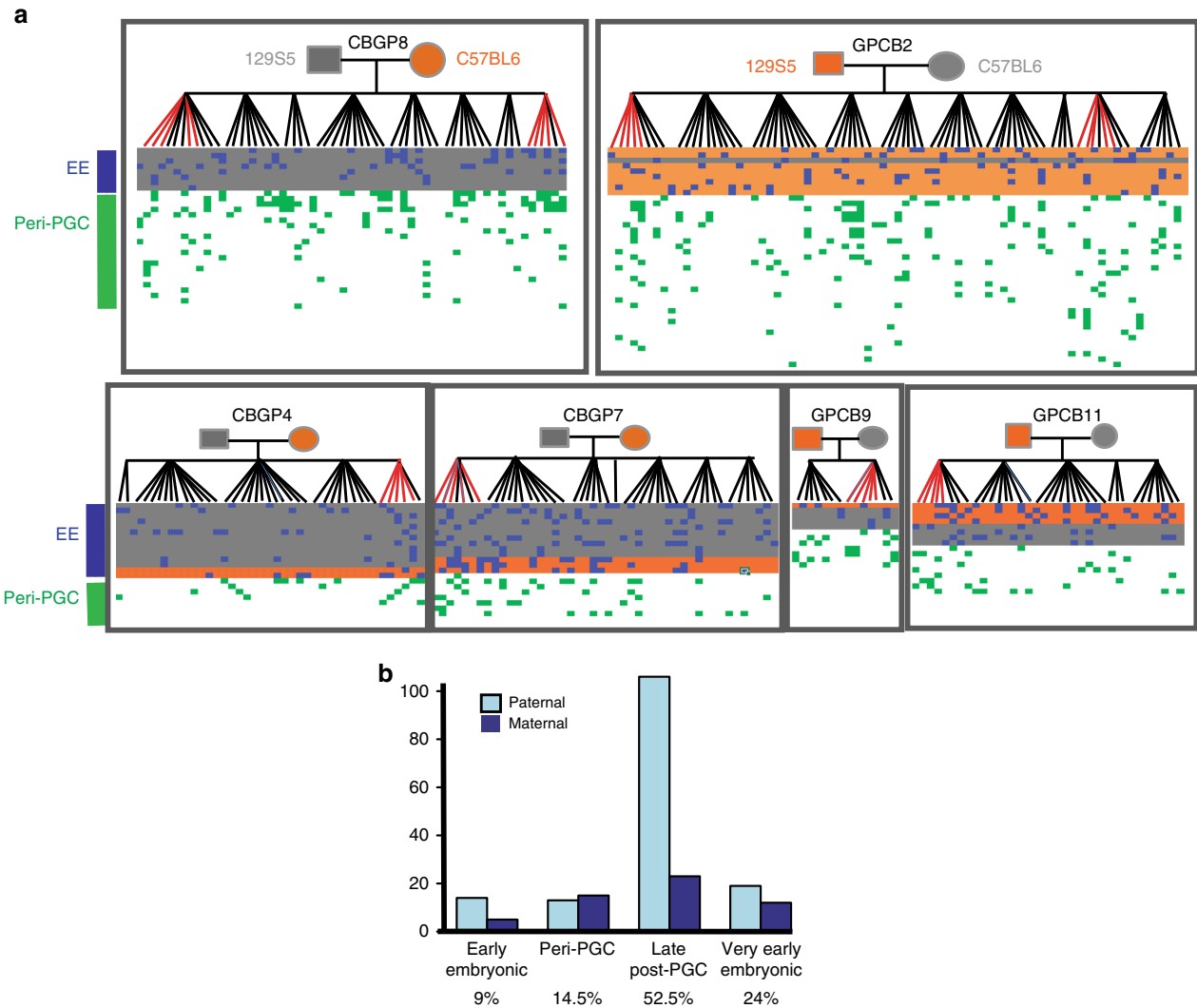

**Fig. 3** Timing of mutations in six mouse pedigrees. **a** Validated de novo mutations in six mouse pedigrees. Offspring are shown vertically, and DNMs are shown horizontally, coloured according to temporal strata. Early embryonic DNMs are shaded according to the parental origin (grey/orange). Late post-PGC and VEE mutations are only observed in one individual and are listed in Supplementary Table 1 and Supplementary Data 1. **b** The number of DNMs assigned to the paternal or maternal haplotype ('Methods') in each temporal strata, with the total contribution of mutations in each strata to the overall rate shown at the bottom of the graph

| Table 1 Autosomal SNV germline mutation rates per generation, per year and per cell division in humans and mice | | |
|---|---|---|
| | **Human** | **Mouse** |
| Mutations per genome per generation | ~71 | ~20 |
| Mutation rate per genome per generation | $1.22 \times 10^{-8}$ [$1.14 \times 10^{-8}$–$1.31 \times 10^{-8}$] | $0.39 \times 10^{-8}$ [$0.37 \times 10^{-8}$–$0.42 \times 10^{-8}$] |
| Mutation rate per year | $4.08 \times 10^{-10}$ [$3.80 \times 10^{-10}$–$4.35 \times 10^{-10}$] | $53 \times 10^{-10}$ [$49 \times 10^{-10}$–$56 \times 10^{-10}$] |
| Average mutation rate per cell division | $5.67 \times 10^{-11}$ [$5.28 \times 10^{-11}$–$6.05 \times 10^{-11}$] | $9.07 \times 10^{-11}$ [$8.41 \times 10^{-11}$–$9.69 \times 10^{-11}$] |

95% confidence intervals calculated assuming Poisson variance around the mean number of mutations ('Methods'). Rate per cell division are calculated assuming that mouse and human generation times are 9 months and 30 years, respectively, and the numbers of cell divisions delineated in ref. [6]. Estimates of mouse mutation rates per year are highly dependent on estimates of average generation time.

reported since the split from the human–mouse common ancestor ~75 million years ago[20,21]. However, comparing the annual mutation rates inferred from the human–chimp and mouse–rat sequence divergence (1.3% and 17%) and age of their most recent common ancestors (~6 MYA and ~12 MYA) suggests a 7-fold difference, only a factor of two different from our estimate[21]. More accurate estimates of average generation times may help to resolve this discordance.

**Mutation spectra in mice and humans**. We observed significant differences ($p = 1.27 \times 10^{-9}$, Chi-squared test) in the mutational spectra in mice and humans[3]. These differences are predominantly characterised by an increased proportion of T > A mutations and a decreased proportion of T > C mutations in mice (Fig. 4a). Mice exhibited a stronger mutational bias towards AT bases than humans (69% vs 59% of all such mutations), in accordance with previous studies that have

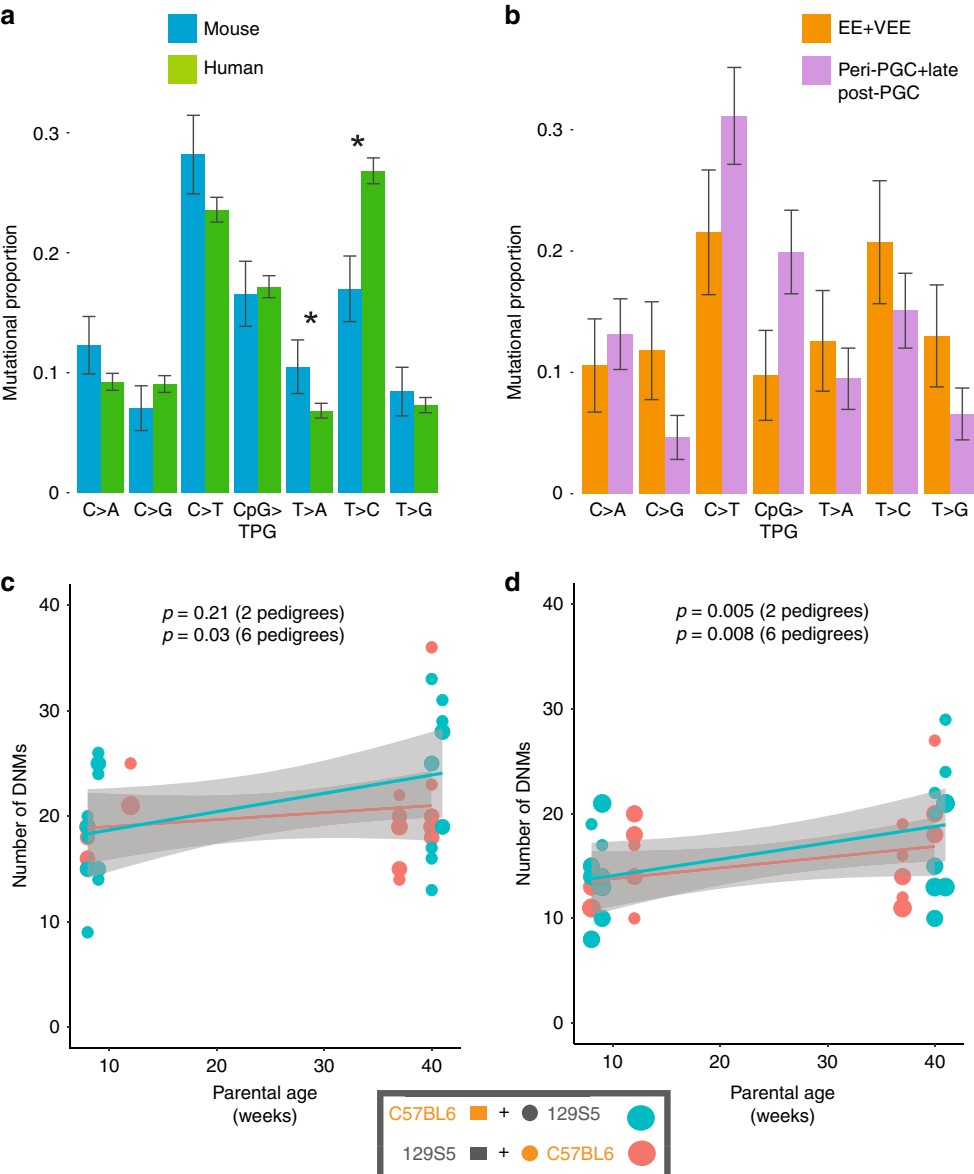

**Fig. 4** Mutational spectra and paternal age effect of DNMs in mice. **a** Comparison of mutational spectra in mice and humans using a catalogue of compiled DNMs in humans[3]. Error bars are 95% confidence intervals. Stars show significance ($p < 0.05$) after correction for multiple testing. The mouse data were derived from DNMs in 40 offspring from six pedigrees and the human data from 6570 DNMs from 109 trios[3]. **b** Comparison of mutational spectra in VEE +EE mutations compared to peri-PGC+late post-PGC mutations in mice. Effect of parental age on number of DNMs in individuals before (**c**) and after (**d**) removal of VEE mutations arising in offspring. The size of the plotting symbol indicates the number of individuals sharing the same number of mutations. $p$ Values indicate the significance of the regression coefficients. The grey shading indicates the confidence limits on the regression. The mouse DNMs were derived from 40 offspring in six pedigrees

suggested that GC content is decreasing more markedly in mouse genomes[22,23]. The differences between the mutation spectra in mouse and humans cannot be accounted for by the slight difference in genome-wide base composition between human and mouse genomes (GC content of 42% and 41%, respectively) as the two most discordant classes of mutation shared the same ancestral base (T) but exhibited opposing directions of change. We observed significant differences ($p = 2.2 \times 10^{-7}$, Chi-squared test) in the murine mutation spectra apparent in earlier (VEE+EE) and later (peri-PGC and late post-PGC) stages of the germline (Fig. 4b), suggesting differences in mutation processes between embryonic development and later gametogenesis. We observed no statistically significant difference between maternal and paternal mutation spectra in mice and humans (Supplementary Fig. 4).

**Parental age effect in mice and humans**. We observed an average increase of 6 DNMs over the 33 weeks between the earliest and the latest mouse litters in the pedigrees where we whole-genome sequenced individuals from the earliest and the latest litters. This is approximately 5-fold greater ($p = 0.0003$, Poisson test) than we would expect in humans over the same time period[1–5]. However, unlike humans, in mice parental age was only a modest predictor of the total number of DNMs per offspring, when data from all six pedigrees are included ($p = 0.03$, linear regression; 'Methods'). We hypothesised that the parental age effect in mice might be obscured by the high proportion of DNMs that represent VEE mutations that arose post-zygotically in offspring and thus would be expected to be unaffected by parental age. Accordingly, we observed a more significant ($p = 0.008$, linear regression) increase in the average number of pre-

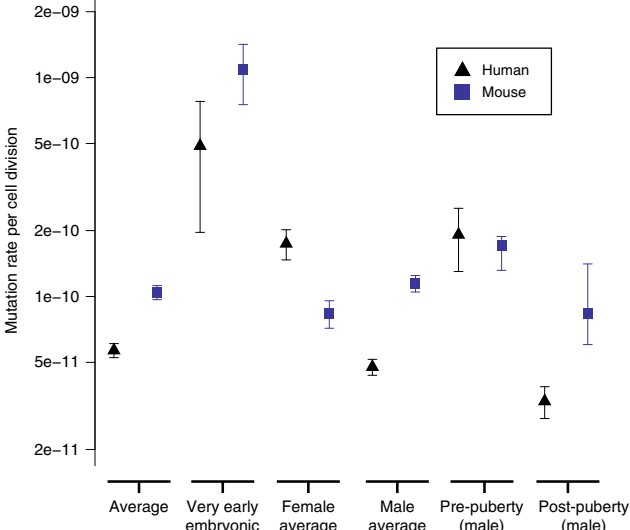

**Fig. 5** Estimation of mutation rates per haploid base per cell division. Mean per-generation mutation rates for SNVs were calculated for mice and from humans using all the data from this study and Rahbari et al.[3] and Kong et al.[1]. The 95% confidence intervals were calculated assuming that DNMs are Poisson distributed, except for the pre- and post-puberty stages, which were derived from the linear models fitted for the paternal age effects, and VEE mutations, which assume quasi-Poisson distribution to allow for over-dispersion. See 'Methods' for further information. The accuracy of the estimates shown here (aside from VEE mutations) rely upon the proposed cellular demographies by Drost and Lee[6]

zygotic mutations (discounting VEE mutations) per offspring with increasing parental age, equating to an increase of ~4.5 mutations per year (Fig. 4c, d). As in humans, the parental age effect in mice is likely to be predominantly paternally driven, as pre-zygotic mutations exhibit the greatest paternal bias (4.3:1 compared to 2.7:1 overall). Humans have a twofold higher turnover in SSCs than mice, and given that the rate mutations accumulate in mice due to parental age is fivefold higher than in humans, this implies that humans have a higher mutation rate per SSC division than mice[6]

**Stage-specific mutation rates in mice and humans**. We calculated mutation rates per cell division at different phases of the germline in humans and mice (Fig. 5), by integrating information on our current understanding of the cellular demography of the germline[6], the paternal age effect, and the numbers of mutations arising in each temporal strata ('Methods'). These estimates do not include uncertainty in the numbers of cell divisions per generation or generation times. Mutation rates per cell division are highest in the first cell division that contributes to the developing embryo in both species. High mutation rates at this earliest stage of embryogenesis is supported by comparable studies in cattle[11]. A high mutation rate during the EE period does not necessarily result in a high germline mutation rate. We inferred the post-puberty mutation rate from the gradient of the paternal age effect. The vast majority of these post-pubertal mutations will have occurred during SSC divisions (especially in humans), and therefore we assume that post-pubertal mutation rates are driven by the mutation rate per SSC division and the number of SSC divisions per year. The most striking difference between the species is the much lower mutation rate in SSC divisions in humans. SSC cell divisions are significantly less mutagenic than all other germline cell divisions in humans but not in mice. SSC divisions account for >85% of all germline cell

divisions in humans but only <40% in mice[6]. We hypothesise that the much greater relative contribution of SSC divisions to the human germline (Fig. 2c) has led to stronger selection pressures to reduce the mutation rate of SSC divisions in humans than in mice.

**Reconstruction of mouse germline genealogies**. Using mutations shared among siblings in the mouse pedigrees, we reconstructed partial cellular genealogies for each parent (Fig. 6, Supplementary Fig. 5). Each parental genealogy is characterised by 2–4 lineages defined by EE and peri-PGC mutations and a residual group of offspring without shared mutations. These primary lineages are distributed randomly with respect to litter timing, suggesting that their relative representation among gametes is stable over time and primarily reflects processes operating prior to PGC specification and/or during the early stages of PGC proliferation. An unknown number of PGC founder cells are specified during mouse embryonic development, which later generate 40–42 PGCs[15–18]. We noted markedly unequal contributions of different parental cellular lineages to gametes (range: 2–53%). The M2 lineage accounted for over half of all gametes and yet was defined by four peri-PGC mutations, suggesting that the fecundity of founding lineages during early PGC proliferation can play a major role. This suggests that specified PGCs do not contribute equally to the final pool of gametes, although further work is required to determine the relative contribution of selective and stochastic factors to the disproportionate representation of cellular lineages among gametes.

### Discussion

The pedigree-based study presented here complements differences in mutation processes inferred from comparative genome analyses[20] but adds sex-specific and temporal granularity. Some of the differences we observed between mice and humans are attributable to the differences in cellular genealogies of the germline, for example, the greater proportion of germline replications occurring during early embryogenesis in mice, leading to greater germline mosaicism and sharing of mutations between siblings. However, other differences represent biological differences within the same stage of embryogenesis or gametogenesis, for example, the striking paternal bias of EE mutations in mice or the reduced rate of mutation in SSC divisions in humans. The cause of this paternal bias in mice perhaps relates to poorly understood, but fundamental, sex differences in how cell lineages are specified in EE development in mice[17,18].

One notable similarity between mouse and human germlines was the hypermutability of the first post-zygotic cell division contributing to the developing embryo (i.e. not including those post-zygotic cells that have undergone cell death or contributed to extra-embryonic tissues), although the relative contribution of these VEE mutations to the mutation rate per generation was much higher in mice. The remarkably high variance in numbers of VEE mutations between mouse offspring suggests that this stage is much more mutagenic for some zygotes than others, though further work is required to characterise the contribution of these mutations to the germline. In addition, reconstructing partial genealogies for the mouse germline has revealed highly unequal contributions of different founding lineages to the ultimate pool of gametes. These observations motivate a deeper understanding of the demography of PGC lineages.

The observation that the mutation rate per cell division in the germline is higher in mice than in humans, despite the mutation rate per generation being lower, accords with a previous study[24]. It has been hypothesised that purifying selection in mice is more efficient at reducing germline mutation rates per generation due

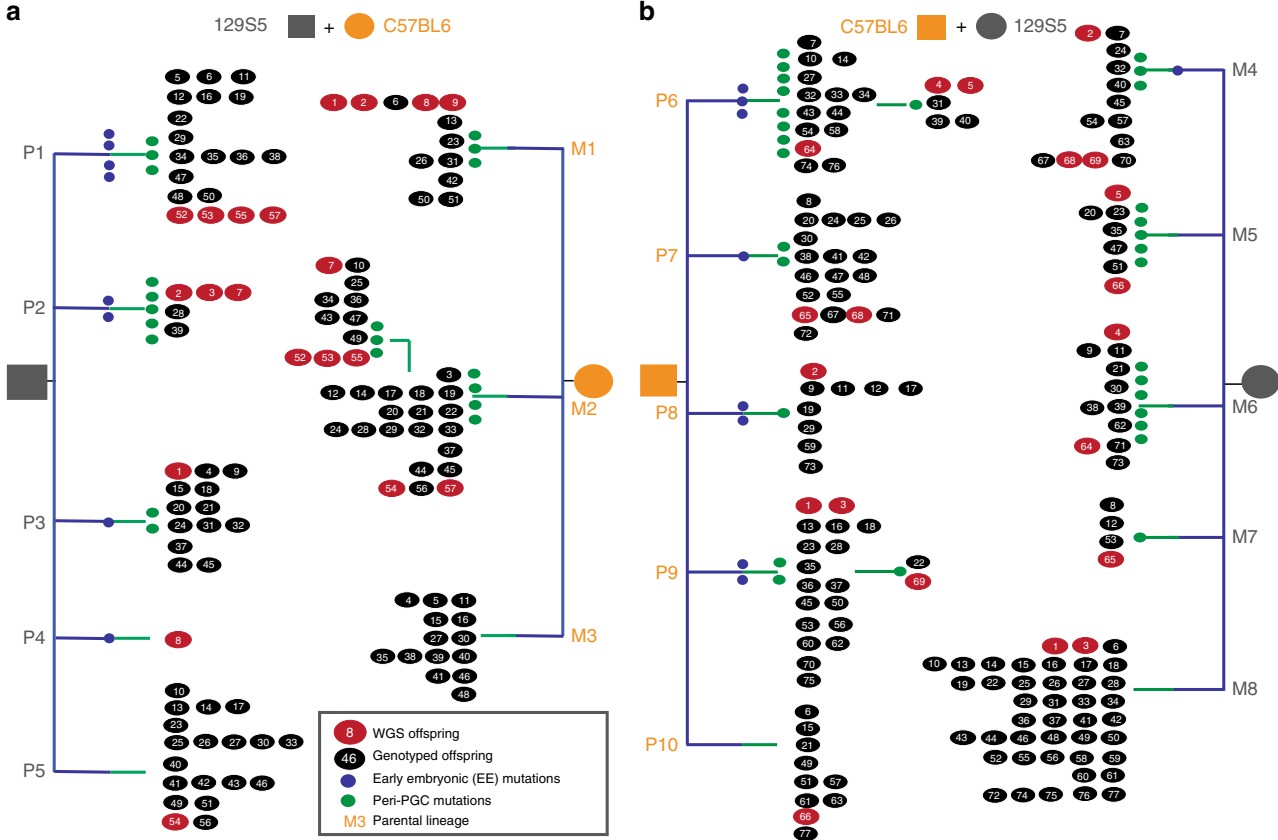

**Fig. 6** Genealogies of maternal and paternal cell lineages delineated by early embryonic and peri-PGC mutations. Parental embryonic lineages reconstructed for each pedigree in the two largest pedigrees (CBGP8 **(a)** /GPCB2**(b)**), as paternal (P) or maternal (M), with each lineage numbered. Mutations delineating the lineages are colour-coded according to their temporal strata as listed in Supplementary Data 1. WGS and genotyped offspring are shown as red and black numbered circles, respectively. Offspring are numbered and ordered according to litter; for example, in lineage P2, individuals 2, 3 and 7 belong to the same litter, whereas numbers 28 and 39 arose in separate, later litters. Lineages P5, M3, M8 and P10 represent offspring without shared mutations and may contain several uncharacterised lineages. Minimal lineages for the four smaller pedigrees can be found in Supplementary Fig. 5

to a larger effective population size[25]. Nonetheless, the selection coefficient of an allele that alters the absolute fidelity of the replication of the genome in every cell division depends on the number of genome replications per generation. Thus, given the much greater number of genome replications in a human generation, an allele that alters the fidelity of genome replication by a certain amount would have a considerably higher selection coefficient in humans than in mice. This factor potentially accounts for the lower mutation rate per cell division in humans despite the likely lower efficiency of purifying selection acting on generational mutation rates.

We hypothesise that the much greater contribution of SSC divisions to the human germline has resulted in a marked reduction in mutation rate per SSC division, both compared to other germline cell divisions in humans and to SSC divisions in mice. Studying germline mutation rates in additional mammalian species with differing proportions of germline cell divisions attributable to SSC will help to test this hypothesis. Recently, the prevalence of dominant developmental disorders in humans caused by DNMs was estimated to be 1 in 300 births[26]. If the mutation rate per SSC division in humans was the same as in mice, the prevalence of dominant developmental disorders would be several times greater.

There are some limitations to this study. First, estimates of cell divisions are subject to the accuracy of the rates reported in Drost and Lee. It would have been beneficial to study patterns of EE and peri-PGC mutations directly in the gonadal tissues of the parents (which were not collected in these mice). Peri-PGC mutations

need to be characterised in ovaries and testes (and their precursor cell lineages) to directly compare the rates and spectra of peri-PGCs between mice and humans. This study is underpowered to detect differences between the spectra of mutations in each strata, due to a limited number of inbred mouse and human pedigrees. It is also possible that, while the mutations rates and spectra we observe are equivalent to those found in other WGS studies, there may be mutation process differences between inbred mice and wild-type (WT) mice. We used reciprocal crosses of both parental strains and saw no strain-specific differences; however, there may be some subtle strain-specific biases due to the differential representation of the two strains in the reference genome, which might become apparent in larger sample sizes. In addition, our sample size is limited to only a few families for each species and so we had limited power to discern between family differences in mutation rates and processes. Lastly, VEEs are typically mosaic in both soma and germline and more work is needed to fully characterise them. Given the detection limits of our study, we are likely to have not been able to detect all the VEE mutations in the offspring.

Recent work has proposed that differences in germline mutation processes between species is dependent on 'life history' traits[27,28]. We contend that these 'life history' traits are imperfect proxies for the true molecular and cellular basis of this variation between species, which relates to the number of different classes of cell division within the germline, and the mutation rates and spectra accompanying each temporal strata of the germline. Pedigree-based studies of other species will establish a truly

comprehensive molecular and cellular understanding of the evolution of mammalian germline mutation processes.

## Methods

**Study design.** The human data are derived from WGS of three healthy families who participated in the Scottish Family Health Study (SFHS). The study was approved by the Human Materials and Data Management Committee (HMDMC) and informed consent was obtained from all participants. The mouse data was derived from WT mice established in the mouse facility at the Wellcome Sanger Institute. The care and use of all mice in this study were in accordance with the UK Home Office regulations, UK Animals (Scientific Procedures) Act of 1986 and were approved by the Wellcome Trust Sanger Institute Ethical Review Committee (AWERB).

**Mice.** Ten male and female mice from two inbred lines (CB57BL/6 and 129S5) were obtained from sib–sib inbred lines previously established at the Wellcome Trust Sanger Institute. Twenty breeding pairs were established: ten CB57BL/6 ♂ × ten 129S5 ♀ (GPCB), and ten of the reciprocal cross: 129S5 ♂ × CB57BL/6 ♀ (CBGP). Breeding pairs were introduced for mating at regular intervals; if a pregnancy resulted, the pups were left to wean and then culled at 3–4 weeks of age. Tissue samples of spleen, kidney and tail were taken from pups and from the parents either when one of them died or became ill or when no pregnancies resulted after matings over a period of 3 months. Once all pairs had stopped breeding, the pair from each cross with the longest breeding span and the most offspring were chosen for the initial WGS experiment. Subsequently, four additional pedigrees, two from each of the reciprocal crosses, were chosen to undertake sequencing in a supplementary sequencing experiment. At the onset of the experiment, the ages of the selected mice were between 8 and 12 weeks (Supplementary Table 1).

**Reanalysis of human pedigrees.** The WGS data and validated DNMs from three multi-sibling trios published in Rahbari et al.[3] were re-analysed. DNMs in humans and mice were discovered and validated separately using comparable pipelines. Average WGS sequence depth was 25× for both the two largest mouse and human pedigrees, and 41× for the smaller mouse pedigrees. The number of DNMs that we discovered in the mice and humans were compatible with our earlier studies and those carried out elsewhere[1,3,13,14]. Classification of EE and VEE mutations in offspring were carried out using the same analytical workflow for both species. The power to classify peri-PGC mutations depends on the power to discover DNMs and the number of offspring sequenced in a pedigree. While the former is comparable between the mouse and human pedigrees, the latter is not. Peri-PGC mutations wrongly classified as late post-PGC mutations are likely to be present in both species, but the human pedigrees are more likely to be affected given the lower number of offspring per pedigree. Detection of DNMs, VEE and EE mutations were carried out using the same pipelines in mice and humans (Supplementary Note 1).

**DNA sample preparation.** Tissues were stored at −80 °C immediately after harvest. DNA was prepared using Qiagen DNeasy kits. Where possible, single DNA aliquots were used for both discovery and validation experiments.

**Sequencing and variant calling.** DNA extracted from the spleen of parents and offspring was sequenced using standard protocols and Illumina HiSeq technologies with a read length of 100 bp in both parents and ten offspring from the earliest and the latest time-matched litters in the two largest mouse pedigrees (GPCB2 and CBGP8). The resultant sequence data were aligned to mouse reference GRCm38. The total mapped coverage after duplicate removal had a mean of 25× and a range of 22–35× for pedigree CBGP8, and 29× and 22–40× for pedigree GPCB2. Variants were called using bcftools and samtools and standard settings[29]. Five offspring and the parents from each pedigree in the remaining four pedigrees CBGP7, CBGP4, GPCB11 and GPCB9 were subject to WGS using standard protocols and Illumina X10 technologies using DNA from the spleen. The total mapped coverage after duplicate removal had a mean of 41× and a range of 41–47× for CBGP4, 38–45× for CBGP7, 39–44× for GPCB11 and 40–42× for GPCB9. Strain and sex specific SNVs were used to confirm the identity of parents.

**DNM calling.** DNMs were called on candidate variants supplied by bcftools using *DeNovoGear* version 0.5 using standard settings[30]. *DeNovoGear* called between 7711 and 11,069 (mean 9736) candidate de novo short indels and SNVs in GPCB2 and CGPB8 and between 6405 and 11,071 candidate SNVs and short indels (mean 9478) in GPCB9, GPCB11, CBGP4 and CBGP7. Candidate de novo SNVs and indels on the X chromosome exhibited marked strain/sex-specific inflation and were discarded. Indels were removed from all analyses.

**Filtering of candidate DNMs.** Candidate DNMs were filtered to exclude simple sequence repeats and segmental duplications, which are sequence contexts highly enriched for false positives. In addition, strain-specific mapping artefacts (low-quality areas leading to clustered/low-quality SNV/indel candidates were filtered by removing sites that had a high variant allele ratio (>20%) in any offspring in the

litter from the reciprocal cross or parent of the reciprocal cross (>4%). Assuming a Poisson distribution for sequencing depth, sites with a depth greater than the 0.0001 quantile were removed due to the likelihood of mapping errors or low complexity repeats introducing false positives. For the two largest pedigrees (CBGP8 and GPCB2), candidate sites where the DNM was present in either parent in >5% of reads and where there were known SNPs in the parental strain were also removed on the grounds that they were likely to be inherited. For the four additional pedigrees (CBGP7, CBGP4, GPCB9 and GPCB11), filtering of candidate DNMs was applied as above but without any upper threshold for the proportion of variant alleles in either parent at a candidate site. Instead, we calculated a mutation-specific error rate at each candidate site using WGS data from all unrelated individuals. Candidate sites that had mutation-specific error rate of >2% in unrelated individuals were removed. Finally, sites with a low variant allele ratio (<15%) in the candidate offspring were removed. Once these filters were applied, 272, 380, 225, 260, 205, 324, 166, 286, 284, 375 and 211, 174, 180, 346, 135, 101, 160, 143, 191 and 300 candidate DNMs remained for GPCB2 and CBGP8 and 61, 65, 74,1 06, 70, 167, 55, 142, 130, 96, 95, 111, 92, 103, 83, 57, 71, 48, 79 and 103 for CBGP7, CBGP4, GPCB11 and GPCB9.

**Experimental validation of DNMs.** We carried out two separate validation experiments, one for pedigrees GPCB2 and CBGP8 and one for CBGP7, CBGP4, GPCB11 and GPCB9. A total of 4460 unique sites across all 20 offspring in GPCB2 and CBGP8 were put forward for validation by targeted sequencing and 1809 unique sites in the CBGP7, CBGP4, GPCB11 and GPCB9 pedigrees (Agilent Sure Select). Twenty-one sites were lost during liftover conversion (conversion from one genome build to another) in the GPCB2 and CBGP8, leading to 4439 sites put forward for bait design. Bait design for both experiments included 2× tiling, moderate repeat masking and maximum boosting, across 100 bp, of sequence flanking the site of interest (extending to 200 bp where baits could not be designed on the initial attempt for both experiments). In GPCB2 and CBGP8, 3253 sites were successfully designed for with high coverage (>50% coverage), 222 with medium coverage (>25% coverage), and 421 with low coverage (<25% coverage). In the 4 additional pedigrees, 1131 sites were successfully designed with high coverage (>50% coverage), 11 with medium coverage (>25% coverage) and 0 sites with low coverage (<25% coverage). Five hundred and sixty-four and 667 sites failed bait design and are likely enriched for false positives. Target enrichment was performed on DNA prepared from tissue samples from parents and offspring using standard protocols and sequenced using 75 bp paired end reads using Illumina Hiseq instruments. We sequenced all candidate DNMs in three tissues from the parents (mean coverage of 400–800× per tissue), two tissues from the WGS offspring (mean coverage of 400×) and a single tissue from all other offspring (mean coverage of 200×). The resultant sequence data were merged by individual and annotated with allele counts at each candidate de novo site using an in-house python script. An in-house R script (http://www.Rproject.org) was then used to calculate the likelihood for each candidate variant being a true DNM, an inherited variant or a false positive call, based on the allele counts of the parents and offspring at that locus. A proportion of the SNV candidates (all sites put forward for validation for one individual) as well as all of the indel candidates were reviewed manually using Integrative Genomics Viewer (IGV)[31]. Sites that were ≤5 bp apart were designated MNVs and counted as one mutation.

**Analysis of mouse pedigree DNM data.** DNMs were called in the two largest mouse pedigrees (CBGP8 and GPCB2) using a pipeline comparable to the human multi-sibling pedigree data described in Rahbari et al.[3]. DNMs in the remaining four pedigrees were called using the same software (see section above) but used improved filtering based on the analyses of the validation data on the two largest pedigrees. Numbers of DNMs observed in mice are not correlated with the coverage or the machine in which they were discovered ($p = 0.2906$, Pearson correlation). There are also no significant differences in mutation spectra ($p < 0.05$) between pedigrees. This demonstrates that the differences between families is minimal and does not impact our conclusions. The lower numbers of offspring subject to WGS in the four smaller pedigrees (5 rather than 10 offspring) means that there is a lower power to observe peri-PGC mutations in these four pedigrees, when compared to the two largest pedigrees. Therefore, we use all six mouse pedigrees for mutation rates, age effect and spectra calculations (where timing is not relevant) and only the two largest pedigrees when comparing the timing of early mutations (EE, peri-PGC) with the human data (Supplementary Note 1).

**Functional annotation of variants.** Functional annotation of validated DNMs was carried out using ANNOVAR[32]. Seventeen DNMs impacted on likely protein function (1 nonsense and 16 missense); however, none were in genes known to have a dominant phenotype in mice or are associated with somatic driver mutations, and so were assumed to be representative of the underlying mutational processes and not under strong selective pressures (Supplementary Table 4).

**Identification of VEE mutations in offspring.** We combined the allele counts across both sequenced tissues in each WGS offspring, after first checking that the allele ratios were concordant across tissues (Fisher's Exact Test). VEE mutations were identified using a likelihood-based test on the combined allele counts by comparing the likelihood of the data assuming the mutation was constitutive

(binomial $p = 0.5$) or was generated in the first cell division giving rise to the embryo (binomial $p = 0.25$). DNMs with an absolute log likelihood difference of >5 were designated as VEE or constitutive. DNMs with an absolute log likelihood difference of <5 was considered unassigned and accounted for 10% of mutations in human pedigrees and 4% in mouse pedigrees.

In addition, phasing of VEE mutations with nearby informative heterozygous SNVs, where available, was used to estimate haplotype occupancy (HO), defined as the proportion of the ancestral haplotype carrying the variant allele. Constitutive mutations should have an HO of 100%, whereas VEE mutations will only be seen on a subset of haplotypes (Supplementary Fig. 6). DNMs shared among offspring (which are constitutive by definition) have high HO, whereas putative VEE mutations were enriched for low HO values that correlated with the estimated level of mosaicism. VEE mutations in mice arose at similar rates in both sexes (average per male offspring = 5, average per female offspring = 6) and approximately equally on paternal and maternal haplotypes (15:9 paternal:maternal) in the largest two pedigrees. Unassigned DNMs were designated as late post-PGC mutations in downstream analyses. VEE mutations in the parents were discovered using a different analysis pipeline described below in the section 'Estimating contribution of VEE mutations to germline mutations'.

**Calculation of haplotype occupancy**. We identified phase-informative heterozygous SNVs with 100 bp of validated DNMs in each offspring. We then determined the phase of the DNM using read-pairs containing both the DNM and the informative heterozygous site. HO is calculated as the proportion of read-pairs that span both the DNM and the informative heterozygous site, from the haplotype on which the DNM arose, that contain the derived DNM allele. DNMs that are pre-embryonic should be in complete phase with one or other of the adjacent alleles (HO = 1). DNMs that arose during embryo development will be observed only on a subset of the ancestral haplotype (HO < 1) (Supplementary Fig. 6).

**Identification and power to detect EE mutations in parents**. EE mutations are observed as DNMs in offspring, which have a statistically significant excess of the mutant allele in one of the parents. In order to identify DNMs that could be mosaic in one of the parents, the site-specific error was calculated for each site (percentage of reads that map to the non-reference variant allele in unrelated individuals from the reciprocal pedigree). This error was then used to calculate the binomial probability of observing $n$ non-reference reads at the mutated site in each tissue in each individual. The probabilities were corrected for multiple testing, using Bonferroni correction, with a threshold of $p < 0.05$ to identify candidate sites, which were then viewed in IGV[31]. In addition, the power to detect mosaicism at different levels (0.5, 1 and 1.5% respectively), in each tissue in each parent was estimated using the sequence depth from the validation data. All EE mutations were observed in all tissues from the mosaic parent.

**Assigning parental origin of EE mutations**. We considered and discounted a wide variety of possible technical artefacts that might explain the apparent parental sex bias we observe in EE mutations in mice. First, sequencing depth, and thus power to detect somatic mosaicism, was equal between maternal and paternal tissues, and the identity of the WGS samples were checked using strain and sex-specific SNVs. Second, where parental origin could be independently determined by phasing with nearby informative sites ($N = 6$), the parental origin was confirmed, thus excluding sample swaps. Third, parental mosaicism in the deep targeted sequencing data was supported by nonzero counts of variant alleles in the WGS data in the corresponding parents at six of the mosaic sites (five paternal, one maternal). Fourth, the same aliquot of DNA was used for WGS and validation by deep targeted sequencing of mutations in parental spleen, lowering the possibility of sample swaps. Lastly, in all cases, parental mosaicism was independently supported by sequencing data from two additional tissues.

**Identification of peri-PGC mutations**. We identified 54 peri-PGC DNMs shared among ≥2 offspring but not present at detectable levels (>1.6% of cells) in parental somatic tissues of the two largest pedigrees (Fig. 3). Peri-PGC mutations arose approximately equally in the paternal and maternal germlines (direct phasing: 6 paternal, 8 maternal; inferred parental origin using co-occurrence: 25 paternal, 25 maternal). We only observed 4 peri-PGC DNMs in the human pedigrees, although the numbers are not comparable between species, due to the disparity in numbers of offspring per pedigree and therefore the power to distinguish this class of DNMs.

**Identification of late post-PGC mutations**. Late post-PGC mutations were observed constitutively in a single offspring and represent mutations arising during cell divisions from PGC proliferation onwards.

**Haplotyping of DNMs in offspring**. We used the read-pair algorithm within *DeNovoGear*[30] to determine the parent of origin of validated DNMs using the whole-genome sequence data. *DeNovoGear* uses information from flanking variants that are not shared between parents to calculate the haplotype on which the mutation arose.

Using this technique, we were able to confidently assign the parental haplotype in 207 of the 753 unique validated DNMs.

**Estimating the autosomal SNV mutation rate per generation**. Sites that were ≤5 bp apart were designated MNVs and counted as one mutation. We estimated the number of autosomal DNMs in each mouse offspring by correcting for the proportion of the genome that was interrogated as follows. Bedtools[33] was used to calculate the proportion of the genome considered in our analysis after removing sites with low- or high-sequence depths for each individual ($p_{depth}$). We then calculated the proportion of sites that were retained after applying our whole-genome filters (simple sequence repeats and segmental duplications) after the depth filters were applied ($p_{filter}$). Last, we used the posterior probability supplied by *DeNovoGear* to calculate what proportion of true DNMs arose at sites that could be validated ($p_{dnm}$). Multisite variants were considered to be a single mutational event. The mutation rate was estimated as follows where $m$ is the average number of mutations observed per offspring.

$$\hat{\mu}_{corrected} = 100 * \frac{m}{p_{depth} p_{filter} p_{dnm}}$$

The percentage of the genome covered after ($p_{depth} p_{filter} p_{dnm}$) ranged from 85.7% to 91.9%, with an average of 89.9%.

To calculate the mutation rates per generation, per year and per cell division for human and mouse comparisons (reported in Table 1), generation times were assumed to be 30 years and 9 months respectively. The average age for the parents in the human study was 29.8 years[3], while the parents in the mouse experiment were aged 24.5 weeks on average when the offspring were conceived. According to Drost[6], generation times of 30 years and 9 months would result in 432 cell divisions in the human germline (401 paternal, 31 maternal) and 87 cell divisions in the mouse (62 paternal, 25 maternal). The mutation rate per generation per haploid transmission was then calculated as follows:

$$\hat{\mu}_{generation} = \left( \frac{\hat{\mu}_{corrected}}{genome \, size} \right) / 2$$

95% confidence intervals were calculated assuming Poisson variance around the mean number of mutations observed. The genome size used was the total number of non-N autosomal sites. The mutation rate per year was then calculated as the mutation rate per generation divided by the average generation time for humans and mice in years.

$$\hat{\mu}_{year} = \frac{\hat{\mu}_{generation}}{generation \, time}$$

Lastly, the mutation rate per cell division was calculated as the mutation rate per generation divided by the sex-averaged number of cell divisions in a generation[6], calculated as the sum of the number of cell divisions in the male and female germline divided by two.

$$\hat{\mu}_{cell \, division} = \frac{\hat{\mu}_{generation}}{N \, cell \, divisions / 2}$$

**Estimation of VEE contribution to germline mutation rate**. We identified candidate VEE mutations that occurred in the EE development of the parents in the deep WGS in mouse parents in four smallest pedigrees. We prioritised candidates in these pedigrees as we had greater power to detect parental VEE mutations due to the greater WGS depth compared to the two larger pedigrees. SNVs were called in each pedigree using Platypus[34]. Raw candidates were filtered to remove sites known to be variant in that mouse strain, sites present in segmental duplications or simple sequence repeats, and retain only those sites called as heterozygous in a single parent.

Candidate sites were then further filtered to retain only sites where the variant base was observed in <5% of the total reads in all individuals in unrelated pedigrees. For each remaining site, the mutation-specific error rate was calculated from individuals in unrelated pedigrees. This mutation-specific error rate was used to remove sites with a Poisson probability of >0.02 of being a sequencing error. Sites with a binomial probability of >0.003 of being constitutive were also removed. Finally, to maximise stringency, we removed candidate sites with >1 alternate read in any other parent, ≤4 variant sequence reads in the candidate individual and where the candidate VAF was >35%.

After applying the filters above, 44 candidate sites remained across the four pedigrees. We validated these mutations across three tissues in both parents and a single tissue in all related offspring using a targeted PCR high-depth indexed sequencing approach[3]. Sites were sequenced (Fig. 1) to an average of 100,000× coverage across the candidate site of interest in the parents and >20,000× in the offspring. We assayed all the available offspring from each pedigree, which increased the numbers of offspring in pedigrees GPCB9 and GPCB11 compared to previous analyses (Fig. 1). The candidate sites were annotated with read counts at the candidate site using an in-house python script. Twenty-six sites out of 29 with high sequence depth in both parents and offspring were classed as true VEE mutations, on the basis of VAF in the parents being incompatible with either sequencing errors or a constitutive variant. For these 26 sites, the number of offspring to whom the mutation was transmitted was determined (Supplementary Fig. 3, Supplementary Table 3).

A linear model was fitted to the outcome of the parental VEE quantification experiment ('Methods', Supplementary Fig. 3, Supplementary Table 3) to infer the relationship between somatic VAF in parents and proportion of offspring carrying the parental VEE mutation. We observed a significant correlation between the VAF of the VEE mutation and the proportion of offspring to which the mutation was transmitted (Pearson correlation = 0.71, $p = 0.00011$; Supplementary Fig. 3, Supplementary Table 3). For simplicity, we used the point estimate from the linear model, rather than a confidence interval. On average, this calculation reduced the contribution of VEE mutations to the germline mutation rate by 40%.

$$\text{Proportion of offspring carrying mutation}_i = \beta_0 + \beta_1 (\text{VAF observed in parents})_i + \epsilon_i$$

This model was then used to predict the likely germline contribution of VEE mutations observed in offspring given their VAF.

$$\text{Adjusted number of VEE mutations in offspring} = 2 \sum_{j=1}^{\# \text{ mutations in offspring}} \left( \beta_0 + \beta_1 (\text{VAF in offspring})_j \right)$$

**Analysis of mutation spectra.** Mutational spectra were derived directly from the reference and alternative (or ancestral and derived) allele at each variant site. The resulting spectra are composed of the relative frequencies of the six distinguishable point mutations (C:G>T:A, T:A>C:G, C:G>A:T, C:G>G:C, T:A>A:T, T:A>G:T), with the C:G>A:T then split into mutations at a CpG and non-CpG context. Significance of the differences between mutational spectra was assessed by comparing the number of the six mutation types in the two spectra by means of a Chi-squared test (df = 6).

**Estimation of SNV mutation rates per base per cell division.** Mutation rates per haploid base per cell division were calculated as described below. De novo indels and de novo SNVs on the X chromosome (in humans) were removed before analysis. MNVs (DNM sites greater or less than 5 bp apart) were counted as a single event. The number of bases at which mutations could have been called was determined on an individual-specific basis by estimating the number of bases accessible to the DNM caller (based on the sequencing depth at each base in each individual) and the number of bases removed by hard filters (simple sequence repeats and segmental duplications). The average number of bases per genome at which mutations could be detected was calculated to be 2,222,635,788 bp in mice and 2,394,138,713 bp in humans. Below, we used the number of mutations per offspring adjusted for the partial contribution of VEE mutations to the germline, as described above. Estimates of cell divisions are subject to the accuracy of the rates reported in Drost and Lee[6].

**Average mutation rates per cell division.** Average mutation rates per base per cell division were calculated by dividing the average number of mutations per offspring by the number of bases in the genome that were interrogated and the average of the total paternal and maternal cell divisions calculated to have occurred in the offspring[6] and divided by two to obtain a haploid mutation rate. The 95% conference intervals were calculated assuming the numbers of mutations per offspring are Poisson distributed.

To calculate the average paternal mutation rate per base per cell division, the mean number of autosomal mutations per offspring was first scaled by the proportion of phased mutations that are of paternal origin, to give the mean number of paternal autosomal mutations per generation, which was then divided by the estimated number of paternal cell divisions per generation (62 in mice, 401 in humans)[6], and the number of bases interrogated in the genome. 95% confidence intervals were derived assuming Poisson variance. The average maternal mutation rate per base per cell division average was calculated similarly, using 25 and 31 cell divisions per generation (mouse and human, respectively)[6].

**Mutation rate per base per cell division for VEE mutations.** VEE mutations occur in the first cell divisions that contribute to the embryo (rather than to extra-embryonic tissues). Modelling (assuming symmetric contributions of daughter cells to the embryo) shows that our mutation calling pipeline on WGS data only had substantial power to detect VEE mutations occurring in the first cell division and therefore present in ~25% of reads and had very low power to detect VEE mutations in subsequent cell divisions (12.5% of reads and lower). Moreover, the distribution of the VAF for VEE mutations is centred symmetrically around 0.25 (Supplementary Fig. 7) as expected for mutations arising in the first cleavage cell division contributing to the embryo. These results suggest that the vast majority of VEE mutations we detected arose in a single cell division.

To estimate the VEE mutation rate per base per cell division, we divided the mean number of VEE mutations per offspring by the number of bases interrogated in the genome and then divided this number by 2 (to obtain a haploid rate). To allow for over-dispersion in VEE mutation rates, we calculated the 95% confidence interval assuming quasi-Poisson distribution[35] in the total number of VEE mutations.

**Mutation rate per cell division pre-puberty (male germline).** The mean number of mutations occurring pre-puberty in the male germline were estimated by subtracting the number of mutations expected to have accumulated since puberty, due to the parental age effect, from the total number of mutations observed in offspring as follows:

$$N = \text{mutations}_{mean} - \left( \text{age}_{mean} - \text{age}_{puberty} \right) \times \text{annual mutations}_{mean}$$

The age at puberty in humans and mice was assumed to be 15 years and 1 month, respectively. The increase in number of mutations per additional year of parental age in humans was taken from the linear model described by Kong et al.[1]. The increase in number of mutations per additional year of parental age in mice was extrapolated from the mean difference (6) in numbers of pre-zygotic mutations between first and last sequenced litters (separated by 33 weeks). The mean number of pre-pubertal mutations per offspring were then divided by the number of bases interrogated in the genome and the expected number of pre-pubertal cell divisions. 95% confidence intervals were derived from the standard error around the slope of the linear model describing the effect of age on the number of DNMs fitted to the data (human data obtained from Kong et al.[1]).

**Mutation rate per base per cell division post-puberty (male).** Mutations accrue in an approximately linear manner with parental age. The post-puberty mutation rate per base per cell division in males was estimated by dividing estimates of the increase in number of mutations per additional year of parental age (see above) by the estimated number of additional SSC divisions per year (42 for mice, 23 for humans[6]) and then by the number of bases interrogated in the genome. Confidence intervals were derived from the standard error around the slope of the linear model describing the effect of paternal age on the number of DNMs fitted to the data (human data obtained from Kong et al.[1]).

**Assessing significance in differences between mutation rates.** Where count data were available, significance between mutation rates per base per cell division was tested using Wilcoxon rank test and Student's $t$ test. Where only mean and confidence intervals were available, significance was tested using the $t$ test only. All tests were adjusted for multiple testing using Bonferroni correction.

**Calculating parental age effect.** We observed an average increase of 6 DNMs over the 33 weeks between the earliest and the latest mouse litters in the pedigrees of the two largest pedigrees. The remaining four pedigrees have information only from one time point in between the earliest and the latest litters (Supplementary Table 1). We therefore combined all six pedigrees and constructed a mixed effects linear model with the pedigree as a random effect to account for differences between pedigrees.

$$N \text{ mutations} \sim (1|\text{pedigree}) + \text{age}$$

**Reconstruction of parental lineages.** For the largest two pedigrees, parental lineages were reconstructed using the distribution of mutations shared between offspring, using an iterative algorithm derived from the following three expectations: This iterative procedure is based on three simple expectations: (i) mutations arising in different parents should co-occur in offspring at random, (ii) mutations present in the same diploid progenitor cell should co-occur in offspring more frequently than expected by chance, and (iii) mutations arising in the same parent but in different progenitor cells should be observed mutually exclusively in offspring ('Methods').

In the first step, mutations belonging to the same parental lineage were identified iteratively using a pairwise test for all pairs of shared mutations, which calculated the binomial probability of $n$ pups sharing $m$ mutations where the frequencies of the mutations were $p$ and $q$ in the offspring. In each step of the iteration, the pair of mutations with the most significant $p$ value were considered to belong to the same parental lineage, as long as the parental origin of the two mutations was not discordant, and were merged into a single 'pseudo-mutation' assigned to all the offspring carrying either mutation. This procedure was then repeated iteratively, with each step involving merging of mutations belonging to the same parental lineage, either until a given $p$ value threshold is reached or the pseudo mutations cannot be merged any further. Using a $p$ value threshold of 0.05, all mutations had completely collapsed into the clusters described. All but 4 of the 70 shared mutations could be assigned to either paternal or maternal lineages.

The accuracy of the lineage reconstruction algorithm was assessed using simulations. First, for each pedigree, shared mutations were randomly re-assigned into the lineages defined by the reconstruction above. The pattern of mutation sharing was then assessed for biological concordance—each offspring can only belong to one paternal and one maternal lineage. The random reassignment of mutations was carried out 10,000 times for each pedigree, but none of these were biologically concordant (i.e. at least one offspring would have more than one paternal or maternal lineage). Second, for each pedigree, mutations were randomly clustered into lineages containing differing numbers of mutations (from 2 to 10 variant sites) and tested again for concordance as above, 10,000 times. In this way, 40,000 simulations across both pedigrees showed no other possible concordant lineage structures. Using this iterative clustering procedure, we assigned 67/70 shared mutations to a specific parent and defined partial cellular genealogies for each parent. These primary lineages are distributed randomly with respect to litter timing, suggesting that their relative representation among gametes is stable over time and primarily reflects processes operating prior to PGC specification and/or during the early stages of PGC proliferation.

Reconstruction of minimal lineages for the four smaller pedigrees was carried out using RAxML version 8.2.12[36]. A matrix consisting of the presence and absence of each peri-PGC and EE mutation in every offspring was constructed for each pedigree. The matrices were split into sites where the parent of origin was known and the RAxML model ASC-BINGAMMA was used with the option –asc-corr = lewis to construct a phylogeny. RAxML reconstructions of the parental lineages of the two largest pedigrees replicated the clades constructed using the algorithm above and shown in Fig. 6.

**Reporting summary**. Further information on research design is available in the Nature Research Reporting Summary linked to this article.

## Data availability

The mouse whole-genome sequences generated as part of this study were deposited in the European Nucleotide Archive (ENA), under study PRJEB1407 and PRJEB14877. The human whole-genome sequencing data[3] are accessible via the European Genome-phenome Archive (EGA) under accession EGAD00001001214. All other data are provided within the paper, in the supplementary materials or are available on request.

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

## Acknowledgements

We are very grateful for the expert assistance of James Bussell and the Sanger Institute Mouse Facility for mouse breeding. We thank Mike Stratton, Inigo Martincorena, Art Wuster, Saeed Al-Turki, Jeremy McRae, Ludmil Alexandrov, Aylwyn Scally, Kirstie Lawson, Ian Adams and Robin Lovell-Badge for thoughtful discussions and sharing of scripts. This work was supported by the Wellcome Trust [grant number WT098051].

## Author contributions

The study was devised by S.J.L. and M.E.H. Sample preparation was performed by S.J.L. Analyses were performed by S.J.L., R.R., J.K., T.K. and M.E.H. The manuscript was written by S.J.L., J.K. and M.E.H. The project was supervised by M.E.H.

## Additional information

**Competing interests:** The authors declare no competing interests.

