## [Peer Review File · Nature Communications]

Reviewers' Comments:

Reviewer #1:

Remarks to the Author:

In "Striking differences in patterns of germline mutation between mice and humans", Lindsay et al. study transmitted somatic and germline mutations and mice. Using extensive sequencing, they assign each mutation into one of four categories: "very early embryonic (VEE), early embryonic (EE), peri-primordial germ cell specification (peri-PGC) and late post primordial germ cell specification (late post-PGC)". The results are compared with analyses of human mutation data that has been generated elsewhere.

The results described here are interesting and provide a meaningful contribution to our understanding of mutation processes in vertebrates.

However, the paper has some serious problems that should be addressed:

1) The writing of the main paper seems overly brief and unstructured. The article does not have a real introduction even though the journal's guide to author's states: "The main text of an Article should begin with an introduction (without heading) of referenced text that expands on the background of the work (some overlap with the abstract is acceptable)". Similarly, I would expect a discussion section that considers the limitations of the approach.

2) The idea of presenting this paper as a comparison of mouse data and human data without showing more than a few summary statistics for the human data seems ill-advised. The contribution of the presented work is the mouse data and the paper should make sure that those results are easily accessible to the reader. The title should also reflect that this is a paper about mouse mutation processes.

3) The 2.5x excess of EE mutations in male mice is interesting, if somewhat implausible. The authors have done some analyses to exclude possible artefacts, but given the extraordinary result, further analyses seem warranted. As the number of mouse families here is limited, it would be for example valuable to assess if the result is driven by one mouse or maybe on strain of mice. The mouse experiment is based on one 1 pair of reciprocal crosses with 10 offspring each and 2 pairs of reciprocal crosses with 5 offspring each. That makes it hard to distinguish gender effects from the effects of the specific crossing; they could even be driven by a single mouse.

4) The methods for assigning mutations into different age categories is creative and provides interesting insights. However, it is very difficult to assess the overall precision of this assignment which is driven among other things by the power to detect heteroplasmies and properties of the individual mice. Given this large uncertainty, the presentation of the results is overly confident, and I doubt that any of the tests for which p-values are presented have the appropriate size. It seems all tests assume that mutations occur independently from each other, when in fact mutations in the same individual are unlikely to be fully independent. Similarly, assuming that most observed VEEs occur in the first cell division is probably oversimplifying, given that while the power to detect specific VEES in later cell divisions is much lower, but there are also bound to be many more of these events. I would expect that these shortcomings are considered in the discussion section of the paper.

.

Minor issues:

In line 418: CGBP7 should be CBGP7.

Line 452: The typical definition of MNV is < 20kb not just <5bp.

The method section clearly indicates that the number of peri-PGC mutations cannot be compared between species (562-564), the main section (line 64-66) suggests that they can be compared. It is not clear how many offspring in the largest mouse families are considered. The main section (line 64-66) suggests a mouse may have up to 19 siblings, suggesting sibships of 20. On the other hand, lines 415-417 and figure 1 only list 10 offspring per family.

P494: The calculation of haplotype occupancy is not clear to me. The paper states " We looked for

heterozygous SNVs with 100bp of validated DNMs in each 496 offspring. We then calculated the phase of the DNM according to the adjacent 497 heterozygous SNV." This does not tell me how phase was calculated.

The modelling of the transmission of VEEs (673-682) is not well motivated. Using a linear model, it makes more sense to use a logistic model rather than a simple regression. Moreover, given that the probability of transmitting a VEE should simply be the VAF, I would like to see the MLEs for beta1 and beta2 and compare them to the VAF; I also find it troublesome that the linear equation only explains ~50% of the variation.

Large and small families have different sequencing technology (Hiseq vs X10) with different coverage properties. That may make it hard to compare or combine them. Number of de novo mutations clearly differs based on the technology used. Is the sequencing technology included as the covariates in the analysis?

Mouse coverage 25x/29x not great for detecting lower level heteroplasmies. This is reflected for example in the observation that only 26/44 VEE candidates were confirmed. Some careful description of detecting power would help interpret the results.

When calculating the number of mutations per cell division, it is not clear that the authors use the appropriate denominator. Given that VEEs, EEs and peri-PGC mutations require a mutation event in an early cell division, I'm not sure that the number of paternal cell divisions per generation (62 in mice, 401 in humans) is the right denominator?

Reviewer #2:

Remarks to the Author:

In this paper, authors presented about 750 de novo germline substitutions as a result of WGS analysis using 40 offspring from 6 pedigrees of mice. This number of the mutations in mice is more than sum of the previous reports, which enable us to know more comprehensive information (parental origin and mutation spectra) about mouse germline de novo substitutions. Remarkable advances in this paper are showing mutation arising timing (temporal strata) and parental age effect in mouse germline, which are novel and important knowledge to understand mammalian germline mutagenesis mechanisms. In my idea, these data support that this paper is worth publishing.

However, I have serious concerns regarding to comparison between the mouse data and human data. Especially, the comparison data of the temporal strata of the DNMs seems to be ambiguous because of the different experimental conditions. The detection capability of DNMs including VEE, EE and peri-PGC mutations would be different between mice and human. For example, the number of sibling is limited in human experiment and sequence data analysis is more difficult in human data because of much more genetic variations in human population than in laboratory mice. This also affects logical reliability about comparison of mutation rates in the SSC divisions. So, I recommend that the authors would revise the manuscript. The paper should show theoretical limitation of the comparisons more clearly and discuss it appropriately. At least, estimates of false negative rate of detection of total DNMs and VEE, EE and peri-PGC mutations in both mouse and human data and the comparison of these values are necessary.

Other specific concerns are shown following.

1. The paper should clarify effects of strain difference (129 and B6) on germline DNMs. If there is substantial difference between 129 and B6, it is not appropriate to combined data of 6 pedigrees directly for statistical analysis. Separate analysis is required as 2 sets of study (father: B6 and father: 129). In our experience, mapping of F1 hybrid mouse NGS data to B6 reference sequence (mm10) decreases some DNMs arise in non-B6 alleles (eg. we can evaluate them by using 129 sequence as a reference for mapping and DNM calls). Estimation and discussion of this type of missing mutations is important for overall discussion in this paper.

2. Previous paper "Differences between germline and somatic mutation rates in humans and mice,

Brandon Milholland et al, 2017, Nature communication”, which shows the per cell division mutation rate and mutation spectra, should be referred appropriately in the manuscript.

3. About line 118, paternal bias among the EE mutations could be one of the most important findings in this study. This result is beyond my anticipation. I think that more conservative statistical analysis should be used than simple two-sided binomial test used in the manuscript. If EE (early embryonic) mutations occurs excessively in developmental process of a limited numbers of fathers by accident, the same results could be obtained. Consistent to this view, Figure 5 exhibits an extremely biased result showing 16 paternal and 1 maternal EE mutations in only 2 sets of parents.

4. About line 175, description of the parental age effects on DNMs should be more clarified. At least, parental age effect (slope) and its confidence interval must be shown in the manuscript. This effect and its reliability are essential for an estimation of mutation rate in SSC divisions in mice. In addition, according to Extended Data Fig 2B, paternal and maternal age effects seems to be different between 2 groups (father: B6 and father: 129). The slopes and p-values of Extended Data Fig 2B should be clarified, too.

5. About line 167, the authors might refer to the evidence that genomic GC content is driven by biased gene conversion to discuss GC content in mouse genome.

6. About lines 391 and 399 and other, the manuscript does not touch with de novo indel data but analysis of indels is described in method section. This contradiction should be resolved.

Reviewer #3:

Remarks to the Author:

Lindsay et al present a unique view of sequence variants segregating in the germline and other tissues of mice. I think their study design and results provides valuable insight into the number of cell lineages segregating in the germline of mice.

However, I am concerned about the anti-conservative nature of their significance tests as these DNMs stem from 6 large sibships with accompanying intra-family correlation. Therefore, statistics derived from the DNM attributes will be correlated within families especially in the case of EE and peri-PGC DNMs.

I find their usage and presentation of the VEE mutations extremely misleading as these mutations do not have to be present in the germline. These mutations should be described and analyzed separately as the authors do not have the offspring of the sibships.

Throughout the manuscript they switch between using the entire set, the two largest pedigrees or the four other pedigrees. It is hard to keep track which subset of the data is being used. Further it seems that they are using different thresholds for the two pedigree sets.

For the general reader it would be helpful to describe the specification of primordial germ cells in the introduction. Especially, as its much better characterized in mice compare to humans (For example Tang et al. Specification and epigenetic programming of the human germ line. Nat Rev Genet 2016; 17(10): 585–600.).

Specific Comments

L18-L20:

The references should be updated and they are missing two recent DNM manuscripts
Goldmann JM et al. Germline de novo mutation clusters arise during oocyte aging in genomic regions with high double-strand-break incidence. *Nat Genet* 2018;50(4):487–92.
Jónsson et al. Parental influence on human germline de novo mutations in 1,548 trios from Iceland. *Nature* 2017;549:519–22.

L41

Please clarify what you mean by unique DNMs. Is it you count DNM once if it is shared among siblings?

L66

Extended table 2 only has the pairwise comparison of the two largest pedigrees. I am not sure what the authors mean by a proportions test in line 66 and the reported p-value is most likely anti-conservative given the numbers in Extended Table 2. More specifically, in Extended Table 2 one family has 5% sharing rate and the other one has 1.49% rate. In the Rahbari manuscript the human sharing values are 0%, 2.3% and 0.3%. Based on the spread of these numbers it is unclear how the significance of the enrichment was derived.

L71-74

Identifying the EE DNMs present in the soma of the parent is a non-trivial task it would be clearer if there would be some description of methodology used for searching for EE DNMs in the main text.

Figure 3.

The germline frequency for a subset of the Peri-PCG DNMs seems to be higher than for the EE DNMs from CBGP8 and GPCB2. However, it is hard to interpret the matrices below the pedigrees as by definition you will only observe the DNM haplotype half of the time and the Peri-PGC DNMs are from both parents. It would be preferential to have a supplementary figure with a version of the matrices presented in Figure 3 with the lineage ordering presented in Figure 5, to see whether the reconstruction of the germ cell lineages is congruent with the somatic presence in the parent.

L114-117

The primordial germ cell specification in mice occurs around gastrulation therefore the frequency across the somatic tissues would be extremely informative for the population dynamics of cells selected to be primordial germ cells. Therefore it would be very beneficial to have this information in Extended Table 3 or portrayed perhaps in a supplementary figure.

L118-L121

It is very counterintuitive that there is a higher paternal bias for the EE DNMs compared to the Peri-PGC DNMs as you would expect the earlier developmental epochs to be similar between the sexes and then diverge later in development. Was the parental DNA processed and sequenced with the offspring DNA? If so then a contamination from the children could result in that Peri-PGC or Late post-PGC DNM would be classified as an EE DNM.

L148-149

The main text describes the VEE DNMs in the offspring as germline DNMs which is inappropriate as the authors do not observe a transmission. However, after reading the methods it seems they are using the VEE DNMs term for the parents rather than the offspring. I am confused as the somatic presence of the DNMs in the parents makes them EE DNMs.

L150

What is the relative contribution of EE DNMs to the mutation rate estimate? Their estimates seem to be comparable to previous mutation estimates introduced in the summary.

L163

Do they also see this species difference in the mutation spectrum of rare and common variants?

L171

The VEE DNMs are not transmitted therefore they should be analyzed separately from the EE, Peri-PGC and Post-PGC DNMs.

Further, is there a mutational spectrum difference between the non-transmitted VEE DNMs and the transmitted EE DNMs?

L175-183

Is the age-related DNM accumulation restricted to the Late post-PGC DNMs? Or do you also see it for the EE and Peri-PGC DNMs?

L214

Why did the authors restrict the lineage analysis to the largest pedigrees? As portrayed in Figure 3 there are plenty of EE and Peri-PCG DNMs in the smaller pedigrees which would allow determination of the cell lineages in those pedigrees.

Figure 5.

The lineages in Figure 5 and in Extended Table 3 do not match. For example, in the CBGP8 pedigree there are two P3 mutations according to Extended Table 3 however there is a single DNM in Figure 5.

If the lineages portrayed in Figure 5 correspond to cellular lineages then the membership of the gametes should be mutually exclusive. However, according to Extended Table 3, the sum of the maximum transmission rate within in each lineage is greater than 50% (24.4%, 24.4% and 9.8%).

Further for GPBP2 the numbering of the lineages does not match and similar for GPBP2 the sum of the maximum transmission rate within in each lineage is greater than 50%.

Overall, this indicates there is something wrong in the lineage designation or the EE/Peri-PGC DNMs are enriched in sequencing artifacts.

L221-223.

The authors could give a lower bound of the number of cells contributing to the founding primordial germ cell population using the number of distinct cell lineages with EE-DNMs.

L428

Not sure what they mean by liftover conversion.

L507-L508

How many sites did they test?

Minor points

- Extra) at line 516
- Extra dot at line 813

Reviewers' comments:

Reviewer #1 (Remarks to the Author):

In “Striking differences in patterns of germline mutation between mice and humans”, Lindsay et al. study transmitted somatic and germline mutations and mice. Using extensive sequencing, they assign each mutation into one of four categories: “very early embryonic (VEE), early embryonic (EE), peri-primordial germ cell specification (peri-PGC) and late post primordial germ cell specification (late post-PGC)”. The results are compared with analyses of human mutation data that has been generated elsewhere.

The results described here are interesting and provide a meaningful contribution to our understanding of mutation processes in vertebrates.

However, the paper has some serious problems that should be addressed:

1) The writing of the main paper seems overly brief and unstructured. The article does not have a real introduction even though the journal’s guide to author’s states: “The main text of an Article should begin with an introduction (without heading) of referenced text that expands on the background of the work (some overlap with the abstract is acceptable)”. Similarly, I would expect a discussion section that considers the limitations of the approach.

We have expanded the initial section to include an introduction section where previous studies on germline mutation rates, and the timing of mutations are discussed. We also expanded the Discussion text to place our analyses in a broader context, and included a discussion on the limitations of the study.

2) The idea of presenting this paper as a comparison of mouse data and human data without showing more than a few summary statistics for the human data seems ill-advised. The contribution of the presented work is the mouse data and the paper should make sure that those results are easily accessible to the reader. The title should also reflect that this is a paper about mouse mutation processes.

We think that the most interesting points are those that contrast between the two species, for example, the differences between EE mutations and the overall mutation rate per cell division in the paternal germline. These differences are only illustrated by comparison between the two species. We have expanded in introduction and discussion to place our work in a wider context. The human work is fully referenced and the data explained in the previous paper. In addition, the following section has been added to the Methods:

Reanalysis of human pedigrees.

The WGS data and validated DNMs from three multi-sibling trios published in Rahbari et al¹³ were re-analysed. DNMs in humans and mice were discovered and validated separately using comparable pipelines. Average WGS sequence depth was 25X and 25X for the two largest mouse and human pedigrees, and 41X for the smaller mouse pedigrees. The number of DNMs that we discovered in the mice and humans were compatible with our earlier studies and those carried out elsewhere¹⁻⁵.

Classification of EE and VEE mutations in offspring were carried out using the same analytical workflow for both species. The power to classify Peri-PGC mutations depends on the power to discover DNMs and the number of offspring sequenced in a pedigree. While the former is comparable between the mouse and human pedigrees, the latter is not. Peri-PGC mutations wrongly classified as Late-Post-PGC mutations are likely to be present in both species, but the human pedigrees are more likely to be affected given the lower number of offspring per pedigree.

We constrained our description of the human data, due to its previously published nature, but if the editor regards it more important to include additional information to aid the reader, then we'd be happy to do so.

3) The 2.5x excess of EE mutations in male mice is interesting, if somewhat implausible. The authors have done some analyses to exclude possible artefacts, but given the extraordinary result, further analyses seem warranted. As the number of mouse families here is limited, it would be for example valuable to assess if the result is driven by one mouse or maybe on strain of mice. The mouse experiment is based on one 1 pair of reciprocal crosses with 10 offspring each and 2 pairs of reciprocal crosses with 5 offspring each. That makes it hard to distinguish gender effects from the effects of the specific crossing; they could even be driven by a single mouse.

We thank the reviewer for this suggestion of further analyses. We evaluated whether there is a family specific effect of EE mutations in the data. To clarify this point, we have added the following lines to the Results:

“We tested the dependence of the EE sex difference on families and found that modelling including an individual effect neither improves a model of parent ~ EE mutations, nor is a significant predictor on its own. (p- value 0.15).”

4) The methods for assigning mutations into different age categories is creative and provides interesting insights. However, it is very difficult to assess the overall precision of this assignment which is driven among other things by the power to detect heteroplasmies and properties of the individual mice. Given this large uncertainty, the presentation of the results is overly confident, and I doubt that any of the tests for which p-values are presented have the appropriate size. It seems all tests assume that mutations occur independently from each other, when in fact mutations in the same individual are unlikely to be fully independent. Similarly, assuming that most observed VEEs occur in the first cell division is probably oversimplifying, given that while the power to detect specific VEES in later cell divisions is much lower, but there are also bound to be many more of these events. I would expect that these shortcomings are considered in the discussion section of the paper.

We do not think our power to detect EE, VEE and Late-Post-PGC mutations will differ substantially between individuals and species. This is because the power to detect EE, VEE and Late-Post-PGC mutations are dependent on WGS sequence depth and the de novo mutation calling pipeline. The mean coverage of the human pedigrees and the largest two mouse pedigrees is 24.7X and 25X coverage

respectively. The smaller mouse pedigrees sequenced have a larger average coverage of 41X. The mouse and human de novo calling pipelines were the same, with minor differences in the filtering strategy of the four smaller mouse pedigrees which are mentioned in the text. All the de novo mutations in this study were sequenced at high depth >200X in at least one tissue in each mouse, and VAFs were consistent between tissues. Therefore, we are confident that the classification of mutations into strata were consistent between individuals and organisms.

We include several analyses that suggest that VEEs occur in a single cell division. To clarify this, we have added the following lines to the text: VEE mutations likely arose within the earliest post-zygotic cell divisions contributing to the developing embryo “and are characterised by a low VAF in the offspring, are absent from parental tissues, with consistent representation in 25-50% of cells across two offspring tissues, reflecting their likely origin within the earliest post-zygotic cell divisions contributing to the developing embryo”.

With regard to the independence of mutations in each individual, we note that the number of mutations per offspring (controlling for parental age) largely follows a Poisson distribution, as expected for random low frequency events. The main departure that we observe from the Poisson distribution of mutations among offspring is specifically for VEE mutations and we specifically highlight this observation as deserving of greater investigation. Our analyses suggest that we are capturing mutations in a full germline cycle and mutations occurring in the one or two cell divisions after VEE mutations will be classified as EE mutations in our study.

Minor issues:

In line 418: CGBP7 should be CBGP7 –

Thank you for noticing this, these typos have been fixed.

Line 452: The typical definition of MNV is < 20kb not just <5bp.

We don't think that the field has yet defined a stable consensus on how to define an MNV. In unpublished analyses (on bioRxiv) we have previously shown that the vast majority of MNVs occur within 5bp and at a separation of 20bp the probability that two alleles reside on the same haplotype decreases to 50% (random). We confine our definition to <5bp simply so as to avoid over-estimating the mutation rate based on clustered mutations occurring at the same time, and do not pursue any other analysis of MNVs. If we were to change our definition as suggested, it would reduce the total number of DNMs in four out of the 20 individuals sequenced by one mutation. We do not think that this would influence our analyses sufficiently to warrant making this change.

The method section clearly indicates that the number of peri-PGC mutations cannot be compared between species (562-564), the main section (line 64-66) suggests that they can be compared.

These are two different analyses. The absolute number of peri-PGC mutations cannot be compared between the two species as their classification is dependent on the number of offspring in each pedigree. However, we can compare the likelihood of any sibling sharing a mutation with any other sibling using a pairwise analysis of mutations shared between sibs vs mutations not shared between sibs. We only use the two largest mouse pedigrees for this analysis as this data shared the same de novo analysis pipeline as the human data. A section to describe this analysis has been added to the methods:

“Estimation of probability of sharing a mutation between siblings.

The probability of an apparent DNM being present in more than one sibling in the same family was calculated as the number of instances of a mutation being shared by two siblings divided by the number of pairwise comparisons between two siblings in all families. Only the two largest mouse pedigrees and the human data were compared as these data shared the same analysis pipelines”

It is not clear how many offspring in the largest mouse families are considered. The main section (line 64-66) suggests a mouse may have up to 19 siblings, suggesting sibships of 20. On the other hand, lines 415-417 and figure 1 only list 10 offspring per family.

The pedigree structures are shown in Figure 1A. The legend describes which of the mice were analysed. Between 5 and 10 mice in the pedigree were WGS. This was followed by genotyping of the DNMs in all the mice in the pedigrees. The line “In the two largest mouse pedigrees, we observed 18% (70/388) of unique DNMs were shared among 2-19 siblings” refers only to the fact that mutations were shared by up to 19 siblings in one pedigree.

P494: The calculation of haplotype occupancy is not clear to me. The paper states “We looked for heterozygous SNVs with 100bp of validated DNMs in each 496 offspring. We then calculated the phase of the DNM according to the adjacent 497 heterozygous SNV.” This does not tell me how phase was calculated.

The phase was determined by using read-pair information to determine which allele at a nearby informative heterozygous site was present on read-pairs containing the DNM. We have edited this paragraph to make this clearer.

“We identified phase-informative heterozygous SNVs with 100bp of validated DNMs in each offspring. We then determined the phase of the DNM using read-pairs containing both the DNM and the informative heterozygous site. Haplotype occupancy is calculated as the proportion of read-pairs that span both the DNM and the informative heterozygous site, from the haplotype on which the DNM arose, that contain the derived DNM allele.”

The modelling of the transmission of VEEs (673-682) is not well motivated. Using a linear model, it makes more sense to use a logistic model rather than a simple

regression. Moreover, given that the probability of transmitting a VEE should simply be the VAF, I would like to see the MLEs for beta1 and beta2 and compare them to the VAF; I also find it troublesome that the linear equation only explains ~50% of the variation.

The reviewer makes an interesting point, but we remain confident that our modelling is appropriate for the following reasons. The VAF that we are measuring is in somatic tissues, whereas the VAF that determines transmission probabilities is the VAF in gametes. Therefore, in this analysis we are estimating the relationship between the VAF in the soma and germline, and we think a linear model is appropriate to reflect this relationship. We assume that the transmission probability is equivalent to the germline VAF, which we think is a reasonable assumption for an allele not under strong selection. We anticipate that independent stochastic processes in the demography of the germline and soma will erode the strength of correlation between germline VAF and somatic VAF, and so we do not anticipate the model to explain all the variance. Moreover, measurement error of germline VAF from finite numbers of offspring also contributes to the variance. In an ideal world we would be able to measure the germline VAF directly, however these tissues are not available to us. We note that using a logistic model, the somatic VAF in the parents is a highly significant predictor of transmission probability ($p=4.78e-14$).

Large and small families have different sequencing technology (Hiseq vs X10) with different coverage properties. That may make it hard to compare or combine them. Number of de novo mutations clearly differs based on the technology used. Is the sequencing technology included as the covariates in the analysis?

The numbers of DNMs are correlated with the age of the parents, as mentioned in the text, and not with the machine/coverage that they were discovered on ($p=0.2906$, Pearson correlation). This is also referred to in the methods section: "Analysis of mouse pedigree DNM data."

Mouse coverage 25x/29x not great for detecting lower level heteroplasmies. This is reflected for example in the observation that only 26/44 VEE candidates were confirmed. Some careful description of detecting power would help interpret the results.

In the manuscript, we use two different methods for detecting VEEs in two distinct analyses, each with their own power limitations. Firstly, we detect VEEs as de novo mutations in offspring from parent trios. Secondly, we detect VEEs in a subset of parents themselves, using careful filtering of sites that are observed as variant in the parent.

In the first analysis, VEEs are detected as DNMs in the offspring and are relatively simple to detect when parental data is available, and where the VAF of the DNM is >12% in the offspring. We use these VEEs in many downstream analyses. The power to detect somatic mosaicism in offspring does decrease rapidly with decreasing VAF, although with the WGS coverage used here the power to detect the class of VEEs that arose in the first cell division remains high.

In the second case, detecting VEEs in the parents without information from the previous generation is not trivial due to the 'background noise' of constitutive variants. However, the inbred nature of the mice meant that with careful filtering, we could capture a subset of this type of mutation. The objective of this experiment was not to capture all the VEEs in the parents, but to estimate the relationship between somatic VAF and germline VAF. We only use these mutations to calculate the transmission probabilities of VEEs detected in somatic tissues. We identified these putative VEEs with lax criteria so as to be confident of validating a sufficient number to perform the linear regression described above. Hence we were not surprised to only validate 26/44.

*We have put a sentence into Identification of Very Early Embryonic mutations in offspring, in the Methods section describing this "VEE mutations in the parents were discovered using a different analysis pipeline described below in the section **Estimating contribution of VEE mutations to germline mutations**" to help clarify this point.*

When calculating the number of mutations per cell division, it is not clear that the authors use the appropriate denominator. Given that VEEs, EEs and peri-PGC mutations require a mutation event in an early cell division, I'm not sure that the number of paternal cell divisions per generation (62 in mice, 401 in humans) is the right denominator?

*We agree that careful thought is needed to apply the correct denominator in these analyses of mutations per cell division. When we calculate the rate based on all the cell divisions that occur in each generation we use the sum of paternal and maternal cell divisions (which is 87 and 432 in mice and humans respectively). We use the number of paternal/maternal cell divisions respectively when we calculate a sex specific rate. This is described in the methods sections **Estimating the autosomal SNV mutation rate per generation** and **Estimation of SNV mutation rates per base per cell division**.*

Reviewer #2 (Remarks to the Author):

In this paper, authors presented about 750 de novo germline substitutions as a result of WGS analysis using 40 offspring from 6 pedigrees of mice. This number of the mutations in mice is more than sum of the previous reports, which enable us to know more comprehensive information (parental origin and mutation spectra) about mouse germline de novo substitutions. Remarkable advances in this paper are showing mutation arising timing (temporal strata) and parental age effect in mouse germline, which are novel and important knowledge to understand mammalian germline mutagenesis mechanisms. In my idea, these data support that this paper is worth publishing.

However, I have serious concerns regarding to comparison between the mouse data and human data. Especially, the comparison data of the temporal strata of the DNMs seems to be ambiguous because of the different experimental conditions. The

detection capability of DNMs including VEE, EE and peri-PGC mutations would be different between mice and human. For example, the number of sibling is limited in human experiment and sequence data analysis is more difficult in human data because of much more genetic variations in human population than in laboratory mice. This also affects logical reliability about comparison of mutation rates in the SSC divisions. So, I recommend that the authors would revise the manuscript. The paper should show theoretical limitation of the comparisons more clearly and discuss it appropriately. At least, estimates of false negative rate of detection of total DNMs and VEE, EE and peri-PGC mutations in both mouse and human data and the comparison of these values are necessary.

We have given careful thought to which metrics are comparable between species and which are likely to be biased due to the inherent differences between the human and mouse data. We agree that peri-PGC mutations cannot be directly compared between the two species due to differences in the number of offspring per pedigree (much lower in humans), which is why we avoid making any between species comparisons for this class of mutation. In contrast, we consider that the VEE and EE mutations are comparable between the two studies, given the individuals were sequenced to a similar coverage and both organisms were analysed using the same pipeline/software. VEE mutations are simply de novo mutations with a low VAF, the detection of which depends on the calling software and sequence coverage for each pedigree. Similarly, EE mutations are de novo mutations with a significant excess of mutant reads in one of the parents. We do not think that there are sufficient differences between the human and mouse studies to influence the conclusions of the study. We have added two sections to the methods: "Reanalysis of human pedigrees" and "Analysis of mouse pedigree data" to expand on how the data from each species was analysed.

With regard to estimates of false negatives: We don't have gold standard data sets in both species against which we can accurately assess our DNM calling and classification errors. We have implicitly corrected for the false negative rate in autosomal DNM detection in our estimation of germline mutation rates by correcting for the numbers of bases that could be effectively interrogated and validated for DNMs (e.g. that pass various filtering criteria, that can be validated), as described in the Methods 'Estimating the autosomal SNV mutation rate per generation'. These corrections are performed consistently across human and mouse and so these mutation rates are comparable.

Other specific concerns are shown following.

1. The paper should clarify effects of strain difference (129 and B6) on germline DNMs. If there is substantial difference between 129 and B6, it is not appropriate to combined data of 6 pedigrees directly for statistical analysis. Separate analysis is required as 2 sets of study (father:B6 and father:129). In our experience, mapping of F1 hybrid mouse NGS data to B6 reference sequence (mm10) decreases some DNMs arise in non-B6 alleles (eg. we can evaluate them by using 129 sequence as a reference for mapping and DNM calls). Estimation and discussion of this type of

missing mutations is important for overall discussion in this paper.

Thank you for this point. We initially designed the experiment hoping to identify strain-specific differences in mutation processes, however, we did not observe any statistically significant differences (e.g. Fig4B) and so combined the data from the two strains together. We have now added this sentence to the Results section:

“We did not observe any significant strain-specific differences between the reciprocal crosses and so combined data from these crosses in downstream analyses.”

We did notice that one class of false positive DNMs was strain-specific and was caused by mapping artefacts. These sites were not biased to any particular offspring and were removed by analysis of the error rate at the putative site in F1 offspring from other pedigrees. We have mentioned also this aspect in the Discussion section on the limitations of this paper.

2. Previous paper “Differences between germline and somatic mutation rates in humans and mice, Brandon Milholland et al, 2017, Nature communication”, which shows the per cell division mutation rate and mutation spectra, should be referred appropriately in the manuscript.

Thank you for suggesting this paper. We have added in a citation in the Discussion, noting that our studies agree that the mutation rate per germline cell division is higher in mice than in humans.

We note that most of the mouse data described in this paper comes from a previous study of ours, but that they recall DNMs using their own pipeline. We also note that they do not attempt exhaustive validation, as we have undertaken here, but rather estimate FDR by validating a subset of sites. We note that the authors estimate a high FDR of ~25%, which accords with the ~30% greater number of putative DNMs that they call compared with the number that we experimentally validated in the same data. Given the differences between the studies, especially with regard to the FDR we think that more quantitative comparisons of mutation rate and spectra are not likely to be informative.

3. About line 118, paternal bias among the EE mutations could be one of the most important findings in this study. This result is beyond my anticipation. I think that more conservative statistical analysis should be used than simple two-sided binomial test used in the manuscript. If EE (early embryonic) mutations occurs excessively in developmental process of a limited numbers of fathers by accident, the same results could be obtained. Consistent to this view, Figure 5 exhibits an extremely biased result showing 16 paternal and 1 maternal EE mutations in only 2 sets of parents.

We were surprised by this finding, and spent many months trying (and failing) to identify possible technical explanations. We have a specific section in the Methods describing the different technical artefacts we explored and discounted ‘Discounting of technical artefacts in assigning parental origin of EE mutations’.

To address the concern about the signal being potentially driven by developmental processes in a limited number of fathers we have formally tested for a significant family effect among the six crosses with respect to the sex-bias in EE mutations, however we detected no significant effect (which accords with the observation by eye in Fig5 that the 4 pedigrees with the greatest number of EE mutations all show a strong paternal bias. We have added the following lines to the Results to show that this observation is not driven by a family effect: “We tested the dependence of the EE sex difference on families and found that modelling including an individual effect neither improves a model of parent ~ EE mutations, nor is a significant predictor on its own. (p- value 0.15).”

4. About line 175, description of the parental age effects on DNMs should be more clarified. At least, parental age effect (slope) and its confidence interval must be shown in the manuscript. This effect and its reliability are essential for an estimation of mutation rate in SSC divisions in mice. In addition, according to Extended Data Fig 2B, paternal and maternal age effects seems to be different between 2 groups (father:B6 and father:129). The slopes and p-values of Extended Data Fig 2B should be clarified, too.

Extended Data Figure 2 has now been moved into the main text. This figure shows the p values for the different regressions, showing that removing VEE mutations increases the significance of the regression. The grey shaded areas show the confidence intervals for the regression, and the strong overlap between the two groups of mouse crosses indicates that we do not have statistical power to detect a strain-specific difference with this number of pedigrees. We have simplified the text to make this section clearer.

5. About line 167, the authors might refer to the evidence that genomic GC content is driven by biased gene conversion to discuss GC content in mouse genome.

This point is referred to in the text in the discussion section. “Mice exhibited a stronger mutational bias towards AT bases than humans (69% vs 59% of all such mutations), in accordance with previous studies that have suggested that GC content is decreasing more markedly in mouse genomes”, and “The differences between the mutation spectra in mouse and humans cannot be accounted for by the slight difference in genome-wide base composition between human and mouse genomes (GC content of 42% and 41% respectively) as the two most discordant classes of mutation shared the same ancestral base (T) but exhibited opposing directions of change.”

6. About lines 391 and 399 and other, the manuscript does not touch with de novo indel data but analysis of indels is described in method section. This contradiction should be resolved.

This was because as standard, DNG outputs short indels as well as SNVs candidate DNMs. This has been clarified by the line “Indels were removed from all analyses”, In the “De novo mutation calling” section of the manuscript.

Reviewer #3 (Remarks to the Author):

Lindsay et al present a unique view of sequence variants segregating in the germline and other tissues of mice. I think their study design and results provides valuable insight into the number of cell lineages segregating in the germline of mice.

However, I am concerned about the anti-conservative nature of their significance tests as these DNMs stem from 6 large sibships with accompanying intra-family correlation. Therefore, statistics derived from the DNM attributes will be correlated within families especially in the case of EE and peri-PGC DNMs.

To address the fact that mutations within families may not be independent, we accounted for a family specific effect in the relevant analyses. We have tested the dependence of the EE sex difference on families and found that modelling including an individual effect neither improves a model of parent ~ EE mutations, or is a significant predictor on its own. (p- value 0.15). Our age effect model incorporates data from all families and accounts for any differences between them. There are also no significant differences in mutation spectra ($P < 0.05$) between pedigrees. This demonstrates that the differences between families is minimal and does not impact our conclusions. We have added a section titled "Analysis of mouse pedigree data" to the Methods to explain more clearly which analyses were performed on which pedigrees and why.

With regard to the independence of mutations within individuals, we note that the number of mutations per offspring (controlling for parental age) largely follows a Poisson distribution, as expected for random low frequency events. The main departure that we observe from the Poisson distribution of mutations among offspring is specifically for VEE mutations and we specifically highlight this observation as deserving of greater investigation in the main text.

We have added the following line to the manuscript "We tested the dependence of the EE sex difference on families and found that modelling including an individual effect neither improves a model of parent ~ EE mutations, or is a significant predictor on its own. (p-value 0.15)."

I find their usage and presentation of the VEE mutations extremely misleading as these mutations do not have to be present in the germline. These mutations should be described and analyzed separately as the authors do not have the offspring of the sibships.

The reviewer is right to say that VEE mutations do not have to be present in the germline. We shared the same concern, which is why we undertook an analysis of parental VEE mutation independent of whether those parental VEE mutations were transmitted to offspring, so that we might estimate the quantitative contribution of

VEE mutations to the germline. In summary, we observed that 21/26 parental VEE mutations were transmitted to offspring, and that the somatic VAF was modestly predictive of transmission probability. These observations were integrated into the estimation of germline mutation rate (e.g. VEE mutations were not counted as 'full' germline mutations, but were weighted by their somatic VAF). These results are shown in Extended Data Figure 2, and are described in the main text.

Throughout the manuscript they switch between using the entire set, the two largest pedigrees or the four other pedigrees. It is hard to keep track which subset of the data is being used. Further it seems that they are using different thresholds for the two pedigree sets.

We are sorry that this was not clearer in the manuscript. We thought a lot about which pedigrees to include in which analyses. We generated and validated the data in the two largest pedigrees first, which then allowed us to refine our DNM filtering criteria for the four other pedigrees to avoid some classes of false positives. When generating the data on the four smaller pedigrees we chose to sequence 5 individuals from four pedigrees rather than a greater number of offspring from a smaller number of pedigrees, so as to maximise the number of EE mutations from different parents and increase our power to determine whether there was significant sex bias in EE mutations.

A section 'Analysis of mouse pedigree DNM data' has been added to the Methods to make clearer which pedigrees were used for which analyses and why:

The following line has also been added to the main text: "We detected 31 mutations as peri-PGC in the four smaller pedigrees, and only observed 4 peri-PGC DNMs in the human pedigrees. The numbers are not directly comparable between species and pedigrees, due to the disparity in numbers of offspring per pedigree and therefore the power to distinguish this class of DNMs."

For the general reader it would be helpful to describe the specification of primordial germ cells in the introduction. Especially, as its much better characterized in mice compare to humans (For example Tang et al. Specification and epigenetic programming of the human germ line. Nat Rev Genet 2016;17(10):585–600.).

The following paragraph, and references therein, has been added to the Introduction:

"Germline mutations can arise at any stage of the cellular lineage from zygote to gamete. Mutations that arise in the first ~10 cell divisions prior to the specification of primordial germ cells (PGCs) can be shared with somatic lineages. In humans, at

least 4% of de novo germline mutations are mosaic in parental somatic tissues¹⁰. Mutations that arise just after PGC specification should lead to germline-specific mosaicism, although the typically small numbers of human offspring per family limit the detection of germline mosaicism, and thus our understanding of mutation processes post-PGC specification. Studies of phenotypic markers in mice have suggested variability in mutation rates and spectra at different stages in the germline, and mutational variability between germline stages has been implicated in recent work in humans, cattle and drosophila.”

We have also added the following sentence to the Results: “Unlike humans, mouse PGC specification is well characterised; after specification, PGCs proliferate rapidly, generating thousands of germ cell progenitors in both sexes^{20,21,22,23}.”

Specific Comments

L18-L20:

The references should be updated and they are missing two recent DNM manuscripts

Goldmann JM et al. Germline de novo mutation clusters arise during oocyte aging in genomic regions with high double-strand-break incidence. Nat Genet 2018;50(4):487–92.

Jónsson et al. Parental influence on human germline de novo mutations in 1,548 trios from Iceland. Nature 2017;549:519–22.

Thank you, These recent references have been included.

L41

Please clarify what you mean by unique DNMs. Is it you count DNM once if it is shared among siblings?

Yes, in this instance, a DNM that is shared among siblings is only counted once.

L66

Extended table 2 only has the pairwise comparison of the two largest pedigrees. I am not sure what the authors mean by a proportions test in line 66 and the reported p-value is most likely anti-conservative given the numbers in Extended Table 2. More

specifically, in Extended Table 2 one family has 5% sharing rate and the other one has 1.49% rate. In the Rahbari manuscript the human sharing values are 0%, 2.3% and 0.3%. Based on the spread of these numbers it is unclear how the significance of the enrichment was derived.

The statistical significance was considered by summing the total number of sib-sib comparisons and instances of sib-sib sharing of DNMs across families within a species, and then performing a proportions test between species. The small number of counts within each family (especially in human pedigrees with smaller numbers of offspring), means that percentage figures within each family will inevitably have high variance, as the reviewer points out. In the revised manuscript, we carry out the analysis for all 6 pedigrees.

A description of this methodology has been added to the Methods.

Estimation of probability of sharing a mutation between siblings.

The probability of an apparent DNM being present in more than one sibling in the same family was calculated as the number of instances of a mutation being shared by two siblings divided by the number of pairwise comparisons between two siblings in all families.

L71-74

Identifying the EE DNMs present in the soma of the parent is a non-trivial task it would be clearer if there would be some description of methodology used for searching for EE DNMs in the main text.

The text introducing EE mutations in the Results has been amended to:

“EE mutations were defined as apparent DNMs observed constitutively in offspring and mosaic in parental somatic tissues, typically mosaic in a lower proportion of cells (2-20%) than VEE mutations consistent with them arising during later embryonic cell divisions, prior to PGC specification (after ~10 cell divisions). “

The sentence “EE mutations are observed as de novo mutations in offspring, that have a statistically significant excess of the mutant allele in one of the parents.” has also been added to the Methods section “Identification and power to detect EE mutations in parents”

Figure 3.

The germline frequency for a subset of the Peri-PCG DNMs seems to be higher than for the EE DNMs from CBGP8 and GPCB2. However, it is hard to interpret the matrices below the pedigrees as by definition you will only observe the DNM haplotype half of the time and the Peri-PCG DNMs are from both parents. It would be preferential to have a supplementary figure with a version of the matrices

presented in Figure 3 with the lineage ordering presented in Figure 5, to see whether the reconstruction of the germ cell lineages is congruent with the somatic presence in the parent.

As CBGP8 and GPCB2 are the largest pedigrees, and 10, rather than 5 offspring were subject to WGS, we have greater power to detect Peri-PGC DNMs in these two pedigrees compared to the four smaller pedigrees, and thus have greatest ability to reconstruct germ cell lineages. The reconstruction of germ cell lineages is 100% congruent with the somatic presence in the parent.

In principle, we think it is not unreasonable that some peri-PGC DNMs might have higher germline frequencies than some EE DNMs, as they may occur only a few cell divisions apart and stochastic demographic factors during PGC proliferation may increase the frequency of some peri-PGC DNMs to above that of some EE DNMs.

We tried generating the figure as you suggest, but could not display all the relevant information in form that we thought would benefit the reader.

L114-117

The primordial germ cell specification in mice occurs around gastrulation therefore the frequency across the somatic tissues would be extremely informative for the population dynamics of cells selected to be primordial germ cells. Therefore it would be very beneficial to have this information in Extended Table 3 or portrayed perhaps in a supplementary figure.

This information has now been added to Extended Table 3, and we have added a Figure (Figure 3C) to show this visually.

L118-L121

It is very counterintuitive that there is a higher paternal bias for the EE DNMs compared to the Peri-PGC DNMs as you would expect the earlier developmental epochs to be similar between the sexes and then diverge later in development. Was the parental DNA processed and sequenced with the offspring DNA? If so then a contamination from the children could result in that Peri-PGC or Late post-PGC DNM would be classified as an EE DNM.

We agree that this finding runs counterintuitive to what might be expected, especially as we do not see a parental sex bias in Peri-PCG mutations. We spent some time thinking about potential technical artefacts, and identified some, but discounted them all. This is described in a dedicated section in the Methods.

“Discounting of technical artefacts in assigning parental origin of EE mutations

We considered and discounted a wide variety of possible technical artefacts that might explain the apparent parental sex bias we observe in early embryonic

mutations in mice. Firstly, sequencing depth, and thus power to detect somatic mosaicism, was equal between maternal and paternal tissues, and the identity of the WGS samples were checked using strain and sex-specific SNVs. Secondly, where parental origin could be independently determined by phasing with nearby informative sites (N=6), the parental origin was confirmed, thus excluding sample swaps. Thirdly, parental mosaicism in the deep targeted sequencing data was supported by nonzero counts of variant alleles in the WGS data in the corresponding parents at six of the mosaic sites (five paternal, one maternal). Fourth, the same aliquot of DNA was used for WGS and validation by deep targeted sequencing of mutations in parental spleen, lowering the possibility of sample swaps. Lastly, in all cases, parental mosaicism was independently supported by sequencing data from two additional tissues.”

Paternal and Maternal DNAs were not processed any differently (with respect to the offspring DNAs), and we note that the EE mutations were observed in all 3 parental somatic tissues, and so each sample would have to have been contaminated to the same degree, which seems highly unlikely.

L148-149

The main text describes the VEE DNMs in the offspring as germline DNMs which is inappropriate as the authors do not observe a transmission. However, after reading the methods it seems they are using the VEE DNMs term for the parents rather than the offspring. I am confused as the somatic presence of the DNMs in the parents makes them EE DNMs.

We concede that the terminologies of the different temporal strata of mutations can be confusing, and have tried to explain this in Figure 2B. In our pedigree data we can potentially detect VEE mutations in both parents and offspring. We can detect VEE mutations most reliably in offspring, and so we base our analyses of the prevalence of VEE mutations on VEE mutations in offspring. However, to investigate the contribution of VEE mutations to the germline we have to study VEE mutations in parents. Therefore there are two separate VEE mutation analyses:

*Firstly, in our WGS and genotyping studies, we observe VEE **in offspring** as DNMs with a VAF of ~12% to ~33%, and EEs in offspring as DNMs that are shared between sibs and observed in parental tissues with VAF <12%. We cannot observe transmission of these are there is no third generation (or unfortunately, material from the gonads of the offspring).*

*However, we carried out additional an experiment, to establish to what degree VEE mutations contribute to the germline. For this, we looked for VEE **in the parental** tissues, without reference to any offspring. This was achieved by standard variant*

*calling in the parents, and filtering to a subset of high-quality variants with a low VAF. Once we found a number of putative VEE mutations in the parents, we then went on to validate these mutations and genotype them in the available offspring. The latter experiment is described in the methods here: **Estimating contribution of VEE mutations to germline mutations.***

We appreciate this part of the manuscript is difficult to describe and would welcome suggestions to improve this.

L150

What is the relative contribution of EE DNMs to the mutation rate estimate? Their estimates seem to be comparable to previous mutation estimates introduced in the summary.

This is an interesting point. The mutation rate calculations assume each EE mutation is an independent event, as even though they only occur once in a parent, on a population level we assume every de novo mutation we observe is independent. We detect 55 EE mutations in our study, but these are calculated as 75 independent events. Given we have 40 offspring in our study, this means that the mutation rate is inflated by ~0.5 of a mutation per mouse offspring, which would not appreciably affect the mutation rate estimates.

L163

Do they also see this species difference in the mutation spectrum of rare and common variants?

This is an interesting question. We and others have shown that mutation spectrum varies with age of allele (due to processes such as biased gene conversion), and that rare (more recent) variants better reflect the true mutation spectrum. Therefore, in order to study this we would need to have population genetic data from wild mice, which is beyond the scope of this study.

We already note that the mutation spectrum differences we observe between humans and mice are consistent with the observation that GC content is decreasing more markedly in rodent lineages.

L171

The VEE DNMs are not transmitted therefore they should be analyzed separately from the EE, Peri-PGC and Post-PGC DNMs.

Further, is there a mutational spectrum difference between the non-transmitted VEE DNMs and the transmitted EE DNMs?

*We show in the Methods section “**Estimating contribution of VEE mutations to germline mutations**” that most (but not all) VEE mutations are transmitted. When estimating mutation rate and paternal age effect, we do indeed treat VEE mutations differently from the other temporal strata to reflect this difference in timing and transmission probability.*

It would be interesting to carry out the mutation spectrum analysis you suggest, however, currently we have insufficient numbers of mutations in each class to detect a meaningful difference. This point is now explicitly mentioned in the Discussion “Secondly, the study is limited to a small number of inbred mouse and human pedigrees, so we are underpowered to detect differences between the spectra of mutations in each strata”

L175-183

Is the age-related DNM accumulation restricted to the Late post-PGC DNMs? Or do you also see it for the EE and Peri-PGC DNMs?

Thank you for this point. We suspect that the age-related accumulation will be strongest in the late post-PGC DNMs, but we simply don't have enough EE and per-PGC mutations to demonstrate this, as we can't distinguish between lack of effect and lack of power.

L214

Why did the authors restrict the lineage analysis to the largest pedigrees? As portrayed in Figure 3 there are plenty of EE and Peri-PCG DNMs in the smaller pedigrees which would allow determination of the cell lineages in those pedigrees.

Unfortunately the smaller number of EE+Peri-PGC mutations, combined with the smaller number of total offspring in the four smaller pedigrees means that the resolution of germ cell lineages is much lower than for the two larger pedigrees, and the computational method applied to the largest pedigrees cannot resolve the lineages. We have used a different method to generate minimal lineages for the smaller pedigrees where (see Methods) and added these to the manuscript (Extended Figure 7).

Figure 5.

The lineages in Figure 5 and in Extended Table 3 do not match. For example, in the CBGP8 pedigree there are two P3 mutations according to Extended Table 3 however there is a single DNM in Figure 5.

If the lineages portrayed in Figure 5 correspond to cellular lineages then the membership of the gametes should be mutually exclusive. However, according to Extended Table 3, the sum of the maximum transmission rate within in each lineage is greater than 50% (24.4%, 24.4% and 9.8%).

Further for GPBP2 the numbering of the lineages does not match and similar for GPBP2 the sum of the maximum transmission rate within in each lineage is greater than 50%.

Overall, this indicates there is something wrong in the lineage designation or the EE/Peri-PGC DNMs are enriched in sequencing artifacts.

Sincere apologies for this error, and thank you for pointing this out. The lineages were not named correctly in Extended Table 3 as according to Figure 6. This has been amended and all entries checked. During this process, we noted that one SNP was place in P8 in error, when it should have been placed in P9. This has also been corrected in Figure 6.

L221-223.

The authors could give a lower bound of the number of cells contributing to the founding primordial germ cell population using the number of distinct cell lineages with EE-DNMs.

This is an interesting idea, and one that we have given some thought to. However, we thought an extended analysis of this kind would require extensive modelling of the complete demography of the germline and so would be out of the scope of this paper, which already contains a complex narrative.

L428

Not sure what they mean by liftover conversion.

Converting sites from one genome build to another. This is now clarified in the text "Twenty-one sites were lost during liftover conversion (conversion from one genome build to another)"

L507-L508

How many sites did they test?

The Bonferroni correction was for the 753 tests of all validated DNMs in all 6 pedigrees.

Minor points

- *Extra) at line 516 - fixed*
- *Extra dot at line 813 - fixed*

Reviewers' Comments:

Reviewer #2:

Remarks to the Author:

Comment to Lindsay et al.

Logical organization in the paper has been improved largely, but I do not think that it is enough for a publication. This paper has both obviously important results and the results that are not sufficient to make some conclusions. However, they seem to be described in almost the same way. In addition, the distinction between novel conclusions and expected conclusions is also ambiguous. I think that it is necessary to rework the composition of the entire manuscript (especially in discussion section) so that the true value of this paper is properly conveyed. Some expert help might be needed for the revision.

Line 120~:

In my previous comments on this paper, I asked for more strict evaluation of false negatives of detected variants. This has several aspects. One of the most important points is how to treat with VEE mutations.

I do not think that the description about VEE mutations is appropriate. Detection of most of VEE mutations exhibiting the variant allele frequency range with 8~32% (values are from Extend Fig 4) from WGS data with 22x coverage are apparently impossible. For example, 10% allele frequency of 20 coverage reads means only 2 reads exhibiting the variant sequence (expected value). And the number of the variant reads follows binomial distribution. Therefore, it is principally impossible to obtain an overall picture about it. In addition, it seems to be meaningless to compare the frequencies of mouse VEE mutations and human VEE mutations. That is because there are distinctive differences in detection power for VEE mutation between in human and in mice (largely depending on a difference between a reference and sequenced data).

The authors should revise the manuscript largely, to make it easier for readers to understand the significance of new findings regarding to VEE mutations. I think that it is logically difficult to discuss VEE mutations and the other germline mutations belonging to the four strata on the same foundation, which are more reliable than VEE mutations. On the other hand, the results that many VEE mutations were detected and their spectrum is different from post-"early embryonic stage" germline mutations seem to be important findings.

Line 174~:

The description about sex bias regarding to EE mutations should be more modest. According to the number of EE mutations shown in Fig 6 and Ex Fig 6, 4 pedigrees exhibited same direction (male > female) but 1 pedigree showed the opposite direction (male < female) and 1 pedigree showed no bias (male = female). I think that this is not sufficient to make a conclusion about sex bias and that this evidence raise the possibility of existence of the sex bias.

Line 269~:

Description about the effect of parental age on the DNMs should be more clarified. This analysis is susceptible to a variance of the number of DNMs. In this analysis, I am afraid that the difference of sequencing coverages of each individual has a potential to affect the parental age effect. So, the authors should add values of sequencing depth and portion of the genome region (pdepth) in each individual on Extended Data Table 1 and show relationship between the sequencing depth and the observed DNM number.

In addition, the authors should show 95% confidence interval in "6 DNMs over the 33 weeks" (Line 269). They had better change "~7 mutations per year" (Line 280) into the value "per 33 weeks" to compare the values easily.

Line 304~306:

Estimation of the number of cell divisions reported by Drost and Lee includes some uncertainty. Therefore, this sentence should be revised.

Line 308~:

Post-puberty mutations which the authors detected and mutations occurred in SSC divisions are quite different in a logical sense. The authors should revise the manuscript to remove this inconsistency.

Line 329:

Add “)” in the end of sentence.

Line 390:

I feel this paradox does not make some sense. The authors should discuss better with reference to the papers such as “Lynch 2010” (cited as number 8 in the manuscript). At least, I did not seem to have any important new ideas presented in this paragraph.

Reviewer #3:

Remarks to the Author:

The authors significantly improved their manuscript and presentation of their results, I believe this study provide valuable insight into the transmission of de novo mutations (DNMs). However, there are still outstanding issues that I would want to see resolved.

I feel that the authors have not addressed my concerns regarding the anti-conservative nature of their tests, especially those concerning the EE and Peri-PGC DNMs. For example looking at the species comparison of the pairwise sharing rate of siblings (L97-L98), they seem to be using a simple chi square test. The p values are most likely underestimated in this comparison due to fact that the pairs of siblings are dependent within a family. This is best demonstrated that you could argue that there is a larger difference (3.5%) between pedigrees CBGP8 and CPCB2 (Fisher’s exact test; Odds=3.48; p-value= $2.1 \cdot 10^{-10}$) than between species with their line of reasoning. This of course incorrect as the effective sample size is much less than the number of sibling pairs.

I used a resampling approach (see the R code below) to derive confidence interval for the difference of sibling sharing between species. The 95% confidence interval for the species difference shows that the sibling sharing rate in mice is comparable to the human estimate. This shows that there is not a striking difference in the sibling sharing rate between the species, therefore I strongly advise that all of the p-values and confidence intervals in the manuscript should account for the intra-family correlation, especially in the case of EE and Peri-PGC DNMs. The authors have not convinced me of the value of using the somatic VEE DNMs in the offspring to quantify the contribution of DNMs that occur in development of parents to the germline mutation rate in mice. This is indirect at best and is based on imputation using the transmission rate of VEE DNMs in a subset of the parents. The VEE DNMs in the children are of interest but portraying them as germline DNMs is misleading, I would strongly recommend to present them separately.

Specific comments:

L269-271 Figure 4:

I see no reason to omit the other 4 pedigrees from this analysis and Figure 4Bii. Although they do not have the same age range as GPCB2 and CBGP8 the 20 offspring in the other 4 pedigrees provide a better estimate of the parental age effect.

Figure 6:

In my opinion this is the most novel and exciting aspect of their work, however, the authors have not addressed my concerns that the germline mosaicism is greater than 50% when all of the cell lineages are combined. From Extended Table 3, the maximum germline mosaic frequency in pedigree CBGP8 are 29.3%, 24.4%, 9.8% and 2.4% (P1, P2, P3 and “private”). As the germline lineages should be mutually exclusive the resulting transmission rate is 65.9% which is greater than the expected 50%. This even clearer in the case of the paternal lineages of GPCB2, the germline mosaic frequencies are 31.7%, 22%, 29.3% and 17.1%, resulting in an overall rate of

100%. As I stated in the last revision, this implies that there is something wrong with the lineage designation or the EE/Peri-PGC are enriched in artifacts. For assessing this it would be beneficial to have the genotype matrices used for the lineage designation as a supplementary table.

Figure 6:

The labeling in Figure 6 for CBGP8 is still discordant with Extended Table 3, I pointed this out in the previous review. The labeling of the CBGP8 lineages is also discordant.

Extended Figure 6:

There is missing annotation of the cell lineages for the smaller pedigrees in Extended Table 3.

Figure 6 and Extended Figure 6 Legend: Is the reference to Supplementary Table 1 correct?

Minor comments:

- Red dot at 54
- Extra dot at 281

R code

```
human<-data.frame(num=c(0,32,2),
dem=c(780,1364,706))
mi<-data.frame(num=c(90,32,22,6,4,4),
dem=c(1800,2151,414,314,424,356))
re<-c()
for (i in 1:10000){
sh<-human[sample.int(3,replace=T),];
sm<-mi[sample.int(6,replace=T),];
re<-c(re, sum(sm$num)/sum(sm$dem)-sum(sh$num)/sum(sh$dem))
}
print(quantile(re,c(0.025,0.975)))
2.5% 97.5%
-0.00353298 0.04067282
```

Reviewer #4:

Remarks to the Author:

This reviewer finds the updated manuscript to be an improvement but major concerns still remain. This is a significant study on germline mutation rates in mice but needs some more modifications before it is suitable for publication.

Major Comments:

1) The readability of manuscript still remains difficult and confusing at places.

The abstract and the title attempt to highlight the differences between human and mice germline mutation rates, but in reality this paper is a significant and thorough study that focuses on Variation in Mutation Rates at Different Stages in the Mice Germline. The human data is too small and not statistically significant (i.e. subdivision of mutations into different categories VEE, Peri/Post PGC) to draw solid conclusions.

The authors have tried to address this concern in the revised manuscript but that does not change reality and I would strongly advise against mentioning this in the title and abstract. Instead the manuscript would significantly improve its readability by focusing on the mice germline mutation rate, and moving the comparison with human mutation rate to the discussion section.

The abstract mentions SSC divisions, but then they are addressed only in page 10 (not even in introduction). As I mentioned before, it maybe best to focus on the germline mutation rate in mice and remove this aspect completely from the abstract.

2) The category of EE mutations is not entirely clear (unlike VEE or Peri/Post PGC variants), and I still cannot understand why they are mosaic in the somatic parental tissue. In other words, I cannot understand how the authors differentiate between the EE and Peri-PGC mutations.

3) The authors haven't really addressed the concerns about lack of statistical power to differentiate between different categories of mutations, and more importantly the assumption behind independence of mutations. I understand that a study of this size will have limitations but this should at least be discussed as a potential challenge.

Minor comments:

1) The abbreviation VAF has been used before it is described in the manuscript.

Reviewers' comments:

Reviewer #2 (Remarks to the Author):

Comment to Lindsay et al.

Logical organization in the paper has been improved largely, but I do not think that it is enough for a publication. This paper has both obviously important results and the results that are not sufficient to make some conclusions. However, they seem to be described in almost the same way. In addition, the distinction between novel conclusions and expected conclusions is also ambiguous. I think that it is necessary to rework the composition of the entire manuscript (especially in discussion section) so that the true value of this paper is properly conveyed. Some expert help might be needed for the revision.

Line 120~:

In my previous comments on this paper, I asked for more strict evaluation of false negatives of detected variants. This has several aspects. One of the most important points is how to treat with VEE mutations.

I do not think that the description about VEE mutations is appropriate. Detection of most of VEE mutations exhibiting the variant allele frequency range with 8~32% (values are from Extend Fig 4) from WGS data with 22x coverage are apparently impossible. For example, 10% allele frequency of 20 coverage reads means only 2 reads exhibiting the variant sequence (expected value). And the number of the variant reads follows binomial distribution. Therefore, it is principally impossible to obtain an overall picture about it. In addition, it seems to be meaningless to compare the frequencies of mouse VEE mutations and human VEE mutations. That is because there are distinctive differences in detection power for VEE mutation between in human and in mice (largely depending on a difference between a reference and sequenced data).

The authors should revise the manuscript largely, to make it easier for readers to understand the significance of new findings regarding to VEE mutations. I think that it is logically difficult to discuss VEE mutations and the other germline mutations belonging to the four strata on the same foundation, which are more reliable than VEE mutations. On the other hand, the results that many VEE mutations were detected and their spectrum is different from post-“early embryonic stage” germline mutations seem to be important findings.

We do not claim to characterise the complete VAF range of VEE mutations in mice and humans for the reason the reviewer suggested. However, given the detection pipeline and sequence coverage are similar between the two species, we think that it is fair to make a comparison between our observations of them in both species, and we have accounted for coverage differences between the species in our analyses (detailed in methods: Estimating the autosomal SNV mutation rate per generation).

Our observation that the mutation rate is high during early embryogenesis has been replicated elsewhere. (Reference 18). We note that we observe more VEE mutations in mice than humans, suggesting that if mapping issues were to make detection harder in mice, our conclusions will increase in validity. We note in the methods that

sequence coverage was in the range of 22-40X for mice and 24.7X coverage (average) in humans, and that calls in mice were concordant between tissues (where multiple tissues were used). We would suggest that the VAF in the mutations shown in Extended data 1 (VAF of VEE and EE observed in different tissues), suggests that we are not missing a huge number of mutations at the higher VAFs relevant to our specific conclusions.

We are able to phase 31 VEE mutations to a parental haplotype, and while the numbers are small, they do not suggest that we have been influenced by the strain of the parents : in CBGP (CB57BL/6 ♂+ 129S5♀ Mother 8: Dad 11), GPCB (129S5 ♂CB57BL/6 ♀ Mother 4: Dad 8). We find a no correlation of between the number of VEEs we find and the depth of sequencing in that offspring. ($r = -0.08$, Pearson's correlation), or the total number of mutations ($r = -0.25$).

We have added a section to the methods to clarify this: "Comparison of Human and Mouse data" Mouse data was analysed using the same pipelines as human. Detection of VEE and EE mutations was carried out using the same pipeline.

We have also added the following line to the discussion of the limitations of the manuscript in the conclusion. Lastly, VEE mutations are typically mosaic in both soma and germline and more work is needed to fully characterise them; given the detection limits of our study we are likely to have not been able to detect all the VEE mutations in the offspring.

Line 174~:

The description about sex bias regarding to EE mutations should be more modest. According to the number of EE mutations shown in Fig 6 and Ex Fig 6, 4 pedigrees exhibited same direction (male > female) but 1 pedigree showed the opposite direction (male < female) and 1 pedigree showed no bias (male = female). I think that this is not sufficient to make a conclusion about sex bias and that this evidence raise the possibility of existence of the sex bias.

We report the facts that there are observable statistically significant differences between mice and humans and the numbers of EE mutations between fathers and mothers in mouse pedigrees. We say that our results are "suggestive of sex differences in the cellular genealogy", and that we "considered and discounted a wide variety of possible technical artefacts that might explain this apparent parental sex bias" which seems reasonable and not an overstatement of the case.

We have added the word "potential" into the following sentence: Further work is required to define these sex-specific differences in germline genealogy, although the observation of early sex dimorphism in pre-implantation murine and bovine embryos^{18,24} may well be relevant to Further work is required to define these potential sex-specific differences in germline genealogy, although the observation of early sex dimorphism in pre-implantation murine and bovine embryos^{18,24} may well be relevant.

Line 269~:

Description about the effect of parental age on the DNMs should be more clarified. This analysis is susceptible to a variance of the number of DNMs. In this analysis, I am afraid that the difference of sequencing coverages of each individual has a potential to affect the parental age effect. So, the authors should add values of sequencing depth and portion of the genome region (pdepth) in each individual on Extended Data Table 1 and show relationship between the sequencing depth and the observed DNM number. In addition, the authors should show 95% confidence interval in “6 DNMs over the 33 weeks” (Line 269). They had better change “~7 mutations per year” (Line 280) into the value “per 33 weeks” to compare the values easily.

The sequence depth for each individual has been added to Extended data table 1. The number of validated DNMs is not correlated with sequence depth ($p=0.1148$, Pearson's correlation test). We harmonised the two statements as the reviewer suggested to the below :

“We observed an average increase of 6 DNMs over the 33 weeks between earliest and latest mouse litters in the pedigrees where we whole genome sequenced individuals from the earliest and latest litters. This is approximately five-fold greater ($p=0.0003$) than we would expect in humans over the same time period ^{2,3,10,11}. However, unlike humans, in mice parental age was only a modest predictor of the total number of DNMs per offspring, when data from all six pedigrees are included ($p=0.03$) (Methods). We hypothesised that the parental age effect in mice might be obscured by the high proportion of DNMs that represent VEE mutations that arose post-zygotically in offspring and thus would be expected to be unaffected by parental age. Accordingly, we observed a more significant ($p=0.008$) increase in the average number of pre-zygotic mutations (discounting VEE mutations) per offspring with increasing parental age, equating to an increase of ~4.5 mutations per year . (Figure 4B)”

Line 304~306:

Estimation of the number of cell divisions reported by Drost and Lee includes some uncertainty. Therefore, this sentence should be revised.

There is no statistical uncertainty in the Drost and Lee paper which can be incorporated into our analysis but we do state this in the paper:

“These estimates do not include uncertainty in the numbers of cell divisions per generation or generation times.” In the legend for Figure 5, we state ” Estimates of cell divisions are subject to the accuracy of the rates reported in Drost and Lee¹⁴.” To make this point clearer, we have now added this sentence to the limitations section :“Firstly, estimates of cell divisions are subject to the accuracy of the rates reported in Drost and Lee”

Line 308~:

Post-puberty mutations which the authors detected and mutations occurred in SSC divisions are quite different in a logical sense. The authors should revise the manuscript to remove this inconsistency.

Context is :“Mutation rates per cell division are highest in the first cell division of embryonic development in both species. High mutation rates at this earliest stage of embryogenesis is supported by comparable studies in cattle.¹⁸ The most striking difference between the species is the much lower mutation rate in SSC divisions in humans. SSC cell divisions are significantly less mutagenic than all other germline cell divisions in humans, but not in mice. SSC divisions account for >85% of all germline cell divisions in humans but only”

We think the reviewer is referring to the fact that some mutations we have classed as Post-PGC will not have occurred during SSC cell divisions but prior to this. This is a valid point. We have added the following text to this section :“ We inferred the post puberty mutation rate; the majority of these mutations will have occurred during SSC divisions (especially in humans), however a small number of mutations in the post puberty class will have occurred prior to SSC divisions.“

Line 329:

Add “)” in the end of sentence.

Thanks, we have fixed this.

Line 390:

I feel this paradox does not make some sense. The authors should discuss better with reference to the papers such as “Lynch 2010” (cited as number 8 in the manuscript). At least, I did not seem to have any important new ideas presented in this paragraph.

Context : The observation that the mutation rate per cell division in the germline is higher in mice than in humans, despite the mutation rate per generation being lower accords with a previous study²⁹. The finding that mutation rates per generation in mice are lower than in humans while per division mutation rates are higher, raises an apparent paradox: if purifying selection in mice is more efficient at reducing mutation rates per generation, why does the murine cellular machinery have, on average, lower fidelity per genome replication? The answer likely lies in the expectation that the selection coefficient of an allele that alters the absolute fidelity of genome replication depends critically on the number of genome replications per generation. Thus, given the much greater number of genome replications in a human generation, an allele that alters the fidelity of genome replication by a certain amount would have a considerably higher selection coefficient in humans than in mice.

In line with the reviewer’s suggestion, this section has been re-drafted as follows:

“The observation that the mutation rate per cell division in the germline is higher in mice than in humans, despite the mutation rate per generation being lower, accords with a previous study²⁹. It has been hypothesised that purifying selection in mice is

more efficient at reducing germline mutation rates per generation due to a larger effective population size (REF 8). Nonetheless, the selection coefficient of an allele that alters the absolute fidelity of the replication of the genome every cell division depends on the number of genome replications per generation. Thus, given the much greater number of genome replications in a human generation, an allele that alters the fidelity of genome replication by a certain amount would have a considerably higher selection coefficient in humans than in mice. This factor potentially accounts for the lower mutation rate per cell division in humans despite the likely lower efficiency of purifying selection acting on generational mutation rates.”

Reviewer #3 (Remarks to the Author):

The authors significantly improved their manuscript and presentation of their results, I believe this study provide valuable insight into the transmission of de novo mutations (DNMs). However, there are still outstanding issues that I would want to see resolved. I feel that the authors have not addressed my concerns regarding the anti-conservative nature of their tests, especially those concerning the EE and Peri-PGC DNMs. For example looking at the species comparison of the pairwise sharing rate of siblings (L97-L98), they seem to be using a simple chi square test. The p values are most likely underestimated in this comparison due to fact that the pairs of siblings are dependent within a family. This is best demonstrated that you could argue that there is a larger difference (3.5%) between pedigrees CBGP8 and CPCB2 (Fisher’s exact test; Odds=3.48; p-value= $2.1 \cdot 10^{-10}$) than between species with their line of reasoning. This of course incorrect as the effective sample size is much less than the number of sibling pairs.

I used a resampling approach (see the R code below) to derive confidence interval for the difference of sibling sharing between species. The 95% confidence interval for the species difference shows that the sibling sharing rate in mice is comparable to the human estimate. This shows that there is not a striking difference in the sibling sharing rate between the species, therefore I strongly advise that all of the p-values and confidence intervals in the manuscript should account for the intra-family correlation, especially in the case of EE and Peri-PGC DNMs.

The authors have not convinced me of the value of using the somatic VEE DNMs in the offspring to quantify the contribution of DNMs that occur in development of parents to the germline mutation rate in mice. This is indirect at best and is based on imputation using the transmission rate of VEE DNMs in a subset of the parents. The VEE DNMs in the children are of interest but portraying them as germline DNMs is misleading, I would strongly recommend to present them separately.

First part refers to : The fraction of mouse DNMs that are shared between two siblings is significantly higher ($p=2.3 \times 10^{-7}$ 97 , proportions test) in mice (2.9%) than has been reported in humans (1.2%)^{13 98} , suggesting that a higher proportion of

DNMs in mice derive from early mutations in the parental germline leading to mosaicism in the parental germline (Extended Data Table 2). We thank the reviewer for this comment and have removed the analysis and changed the sentence as follows : “This observation suggests that an appreciable proportion of DNMs in mice derive from early mutations (and therefore germline mosaicism) in the parental germline.” We have removed the relevant Extended Data table and methods section from the text.

We have shown that VEE mutations arise de novo during early embryonic development, and a high proportion of them are transmitted to the next generation. See methods section “Estimating contribution of VEE mutations to germline mutations”, and “Extended Data Figure 4: Germline mosaicism of VEE mutations observed in the parents.”, where we show that the VAF of a VEE mutation has a broadly linear relationship with germline mosaicism (proportion of offspring that have inherited the VEE mutation that arose de novo in the parent). However, we agree that there are not “germline” mutations in the traditional sense, as they arise before lineage specification of the germline and soma. In order to make this distinction clearer, we have added the following sentence to the early part of the text where we describe our strata“ As VEE mutations are detected in the offspring, and arise in the embryo before the lineages for the germline and soma are specified, so in principle VEE mutations can be restricted to the germline or soma, or can be shared by both.” In addition, we have added the caveat “The high mutation rate during the early embryonic period is not necessarily indicative of a high germline mutation rate”to the section where we model mutation rates at different stages. We have also changed “The remarkably high variance in numbers of VEE mutations between mouse offspring suggests that this stage is much more mutagenic for some zygotes than others” to “The remarkably high variance in numbers of VEE mutations between mouse offspring suggests that this stage is much more mutagenic for some zygotes than others, though further work is required to characterise the contribution of these mutations to the germline,” in the conclusion.

Specific comments:

L269-271 Figure 4:

I see no reason to omit the other 4 pedigrees from this analysis and Figure 4Bii. Although they do not have the same age range as GPCB2 and CBGP8 the 20 offspring in the other 4 pedigrees provide a better estimate of the parental age effect.

All six pedigrees are included in this analysis. This is mentioned in the methods : “Calculating Parental age effect”

We therefore combined all six pedigrees and constructed a mixed effects-linear model with the pedigree as a random effect to account for differences between pedigrees.

Figure 6:

In my opinion this is the most novel and exciting aspect of their work, however, the authors have not addressed my concerns that the germline mosaicism is greater than 50% when all of the cell lineages are combined. From Extended Table 3, the maximum germline mosaic frequency in pedigree CBGP8 are 29.3%, 24.4%, 9.8% and 2.4% (P1, P2, P3 and “private”). As the germline lineages should be mutually exclusive the resulting transmission rate is 65.9% which is greater than the expected 50%. This even clearer in the case of the paternal lineages of GPCB2, the germline mosaic frequencies are 31.7%, 22%, 29.3% and 17.1%, resulting in an overall rate of 100%. As I stated in the last revision, this implies that there is something wrong with the lineage designation or the EE/Peri-PGC are enriched in artifacts. For assessing this it would be beneficial to have the genotype matrices used for the lineage designation as a supplementary table.

We think the reviewer has misunderstood the nature of this analysis. These figures show placement of the offspring onto hypothesised parental (diploid) lineages, so all offspring should be present once on a single lineage. The lineages are derived from de novo mutations in the parent shared among offspring: As the offspring represent a haploid sampling of a diploid lineage, not all of the offspring assigned to a particular lineage will have all the mutations that define the lineage. On average, each offspring will have 50% of the mutations defining their ancestral diploid lineage.

This method is also used to reconstruct human lineages in a similar manner in : Multiple transmissions of de novo mutations in families *Nature Genetics* volume 50, 1674–1680 (2018). In the latest version of the manuscript we use the same method (RAxML) in the *Nature Genetics* paper to recapitulate our mouse lineages (reference 41). In the methods section “Reconstruction of parental lineages” we state the following:

“Reconstruction of lineages for the four smaller pedigrees was carried out using RAxML version 8.2.1241. A matrix consisting of the presence and absence of each Peri-PGC and EE mutation in every offspring was constructed for each pedigree. The matrices were split into sites where the parent of origin was known and the RAxML model ASC-BINGAMMA was used with the option `–asc-corr=lewis` to construct a phylogeny. RAxML reconstructions of the parental lineages of the two largest pedigrees replicated the clades constructed using the algorithm above and shown in Figure 6.”

Figure 6:

The labeling in Figure 6 for CBGP8 is still discordant with Extended Table 3, I pointed this out in the previous review. The labeling of the CBGP8 lineages is also discordant.

Thankyou for pointing this out, the table and the figure have been checked, and are now concordant.

Extended Figure 6:

There is missing annotation of the cell lineages for the smaller pedigrees in Extended Table 3.

Thankyou for this comment, we couldn't find a missing label but would be happy to fix this if the reviewer could instruct us further.

Figure 6 and Extended Figure 6 Legend: Is the reference to Supplementary Table 1 correct?

Thankyou for this observation, this is fixed.

Minor comments:

- Red dot at 54
- Extra dot at 281

R code

```
human<-data.frame(num=c(0,32,2),
dem=c(780,1364,706))
mi<-data.frame(num=c(90,32,22,6,4,4),
dem=c(1800,2151,414,314,424,356))
re<-c()
for (i in 1:10000){
sh<-human[sample.int(3,replace=T),];
sm<-mi[sample.int(6,replace=T),];
re<-c(re, sum(sm$num)/sum(sm$dem)-sum(sh$num)/sum(sh$dem))
}
print(quantile(re,c(0.025,0.975)))
2.5% 97.5%
-0.00353298 0.04067282
```

Reviewer #4 (Remarks to the Author):

This reviewer finds the updated manuscript to be an improvement but major concerns still remain. This is a significant study on germline mutation rates in mice but needs some more modifications before it is suitable for publication.

Major Comments:

1) The readability of manuscript still remains difficult and confusing at places. The abstract and the title attempt to highlights the differences between human and mice germline mutation rates, but in reality this paper is a significant and thorough study that focuses on Variation in Mutation Rates at Different Stages in the Mice Germline. The human

data is too small and not statistically significant (i.e subdivision of mutations into different categories VEE, Peri/Post PGC) to draw solid conclusions.

The authors have tried to address this concern in the revised manuscript but that does not change reality and I would strongly advise against mentioning this in the title and abstract. Instead the manuscript would significantly improve its readability by focusing on the mice germline mutation rate, and moving the comparison with human mutation rate to the discussion section.

The abstract mentions SSC divisions, but then they are addressed only in page 10 (not even in introduction). As I mentioned before, it maybe best to focus on the germline mutation rate in mice and remove this aspect completely from the abstract.

2) The category of EE mutations is not entirely clear (unlike VEE or Peri/Post PGC variants), and I still cannot understand why they are mosaic in the somatic parental tissue. In other words, I cannot understand how the authors differentiate between the EE and Peri-PGC mutations.

We respectfully largely disagree with the reviewer when they state that the “The human data is too small and not statistically significant” as despite the small sample size, we show that there are clear differences between mutation rates at different stages. In the revised manuscript, we have removed one comparative analysis of human and mice data, as described above, which was not statistically significant on re-analysis. However, having reviewed all the other tests we feel these are robust and support appropriately described conclusions. We have added a section to the methods (in “Reanalysis of human pedigrees”, to make it clear that both species have comparable sequence coverage and were analysed using the same pipelines.” Detection of DNMs, VEE and EE mutations were carried out using the same pipelines in mice and humans.”

Given the need to extensively reanalyse the published human data to make it comparable to the mouse data, and the inclusion of comparative analyses between species in most of the Results sections we do not feel moving the comparisons to the Discussion would be a fair reflection of the contribution of the comparative analyses throughout the manuscript. Nonetheless, if the editor feels that the title proposed by the Reviewer (‘Variation in Mutation Rates at Different Stages in the Mice Germline’) better captures the novelty in the manuscript then we would be happy to change the title to this, or similar.

We welcome the Reviewer’s suggestion to improve the readability of the manuscript. This manuscript comprises work that is inherently difficult to describe given the imperfect mapping of detectable temporal stages of mutation onto the biology of the germline. This is especially true given the somewhat immature state of the currently used terminology to describe mutations with different temporal origins (e.g. ‘somatic’ and ‘germline’ mosaicism, which are used variably in the literature). We have gone to some lengths to try to introduce a more granular and hopefully helpful data-driven classification of the temporal origin of mutations. The field clearly needs new nomenclature to capture greater biological granularity and distinguish between data-defined and biologically defined temporal stages. However, we feel that the observation that the mutation rate in SSC divisions is very different between mice and

humans is one of the most substantive conclusions of our manuscript and is robustly supported by the data, so we would prefer to keep this conclusion in the abstract. We have added a sentence to the Introduction on SSCs (in bold):

“Germline mutations can arise at any stage of the cellular lineage from zygote to gamete. Spermatogonial Stem Cell (SSC) divisions in post-pubertal males account for the highest proportion of all cell divisions in the germline. Mutations that arise in the first ~10 cell divisions prior to the specification of primordial germ cells (PGCs) can be shared with somatic lineages.”

We would welcome further suggestions from the Editor to improve the readability of the manuscript, given the nomenclature challenges described above.,

We have added the following lines to the text to clarify the uncertainty around the rate of VEE mutation and their contribution to the germline.

“As VEE mutations are detected in the offspring, and arise in the embryo before the lineages for the germline and soma are specified, so in principle VEE mutations could be restricted to the germline or soma, or could be shared by both.”

“Lastly, VEE mutations are typically mosaic in both soma and germline and more work is needed to fully characterise them given the detection limits of our study we are likely to have not been able to detect all the VEE mutations in the offspring.”

EE and Peri-PCG mutations are different in that EE mutations are detectably mosaic in the parental germline and soma, whereas Peri-PGC mutations are detectably mosaic only in the parental germline, which reflects the timing of the mutation (ie before or after the split of germline/somatic lineages, or that Peri-PCG mutations are somatically mosaic below our detection level.) We have tried to illustrate this in Figure 2A and describe it in the text. We find that there are differences between mosaic mutations that occur in these strata, EE mutations have an apparent sex bias in mice, whereas Peri-PGC mutations do not.

3) The authors haven't really addressed the concerns about lack of statistical power to differentiate between different categories of mutations, and more importantly the assumption behind independence of mutations. I understand that a study of this size will have limitations but this should at least be discussed as a potential challenge.

With regards to the assumption behind independence of mutations, we have addressed this in several ways. Firstly, within individuals we have treated clustered mutations (within 5bp) to represent single mutational events in our analyses. Beyond that the number of mutations per offspring largely follow a poisson distribution, which would not be expected if there were major residual non-independence between mutations. Secondly we have accounted for family specific effects, which may affect the independence of mutations, by using mixed effects models in the relevant analyses. To reflect the reviewers concerns, we have added the following statement to

the Discussion in the section focusing on limitations of our study “In addition, our sample size is limited to only a few families for each species and so we had limited power to discern between family differences in mutation rates and processes.”

Minor comments:

1) The abbreviation VAF has been used before it is described in the manuscript.

Thanks for this observation, this has been corrected.

Reviewers' Comments:

Reviewer #2:

Remarks to the Author:

Comment to Lindsay et al.

The authors improved their manuscript well. I think the presentation provides new insight into the de novo germline mutations in mice, which merit publication. However, there remain several issues which the authors have to treat with.

Issues:

Line 149-151 and Line 154-157:

Revise this correctly. The same sentence is repeated twice.

Line 263:

Remove “)”

Line 267:

“We observed no difference between ~”

I think that Ex-data Fig6 shows potential difference in the spectra, albeit statistically no difference. So, the author should clarify the sentence. For example, “we found no statistically difference at least within our detection limit”.

Line 284-288:

Please clarify this sentence.

Line 307 and other place:

It is doubtful that mutation rate per cell division is the highest in the first cell division. The authors should remark the possibility that many of early embryonic cells would not contribute epiblast. If most of early embryonic cells lead to trophoblast or cell death, the discussion would be completely different. The authors should revise the manuscript to clarify this point.

Line 458 and Line 535:

The same paper is listed twice as a reference.

Line 794-:

I cannot understand the following formula to estimate per generation mutation rate.

$u = m / p_{\text{depth}} p_{\text{filter}} p_{\text{dnm}}$

I think that the mutation rate cannot be obtained by dividing the average number of mutations by percentages. Please clarify this. In addition, the authors should show all values (including 3 Ps) which were used in this study. These values must be helpful for readers to understand the calculation process and the accuracy of this analysis.

Reviewer #3:

Remarks to the Author:

The study design is novel and provides insight into where and when mutations occur between subsequent generations. The authors have improved presentation of their results and highlighted the difference between VEE mutations and the transmitted ones. However, I find that the authors have partially addressed my concerns regarding the presentation of the VEE mutations in the text. I realize, this is a non-trivial task to introduce these concepts to the general reader, however, here are some suggestions that could help the general reader to understand the distinction between the VEE mutations and the other classes, i.e. they most likely occurred in the offspring but not the parent.

In the temporal strata column in Fig 2A, be explicit in the text that EE are constitutional in offspring, mosaic in the parents, and so on for the other classes. Maybe add a dashed line to 2A, separating the VEE mutations from the other mutations. In figure 2B, the relationship of inner/outer circle to parent/child should be annotated.

The authors insist incorporating the VEE mutations into the mutation rate analysis, they claim their imputation models their transmission rate reasonably well. However, given the non-transmitted nature of the VEE mutations I would want to see an extra column in Table 1 with the mutation rates calculated without the VEE mutations.

Specific Comments

Lines 149-157: The VEE mutations are discovered by comparing them to binomial with $p=0.25$ and they were required to be concordant across tissues according to the methods (Lines 704-709). Therefore, I find that they are describing their criteria of detecting VEE mutations rather than novel results.

Figure 6. I understand that the authors are reconstructing the parental germ cell lineages (diploid) through the haploid representation of the offspring. Furthermore, I understand that if the offspring inherits the non-mutated allele at the mutated loci, the offspring is not necessarily a carrier of mutation defining the diploid cell lineage. The authors define germline mosaic frequency (GMF) like this "Germline mosaic frequency was calculated as number of offspring carrying mutation/total number of offspring assayed" in the caption of Extended Data Table 2. Then a mutation present in all germ cells of the parent should have an expected GMF value of 50%. In light of this, it is improbable to have GMFs of 31.7% (P1; 62.4% of germ cells), 22% (P2; 44% of germ cells), 29.3% (P3; 58.6% of germ cells) and 17.1% (P4; 34.2% of germ cells) for mutually exclusive cell lineages from GPCB2 (father), as the sum is greater than 100% (200% of germ cells). These values indicate that these cell lineages are not mutually exclusive and therefore the topology of the cell lineages reported in Figure 6 should be revised. Is there perhaps a discrepancy how the GMF is calculated and how it is defined? For example for chr1: 195105306:C>T, the number of carriers is 8 out of 77 genotyped (GMF of 10.3%) but the reported GMF in Extended Data Table 2 is 19.5%.

Minor comments

Line 52: Is this the correct reference (14)? I would guess the correct reference is Rahbari et al 2016. This work should also be referenced "Multiple transmissions of de novo mutations in families" Nature Genetics volume 50, 1674–1680 (2018).

Line 66: extra space after the citation.

Line 80: There is no reference to Figure 2c in the text, maybe it could be referenced here

Figure 1 caption: between 41 and 77 offspring, should be probably be 14 and 77 offspring.

Lines 136-144: perhaps incorporate this into the introduction?

Figure 4: the images are in low resolution.

Figure 4: it would be nice to have the number of families and offspring in each panel.

Line 396: the reference is not in the correct format.

Reviewer #4:

Remarks to the Author:

I find the manuscript to be a marginal improvement (reads slightly better with some additional analyses) but it seems we have reached an impasse about my major concern from last time i.e. the comparison with human mutational data.

I still think that this is a significant study on germline mutation rates in mice, but the weak and underpowered comparisons with human data do not merit the current title or the conclusions highlighted in the abstract. This point has been raised by the other reviewer too, who clearly sees

the scientific merit in this manuscript, but gets distracted by the novel vs expected conclusions.
I agree with the authors suggestion that the editor should make the final call about this.

Reviewers Comments

Reviewer #2 (Remarks to the Author):

Comment to Lindsay et al.

The authors improved their manuscript well. I think the presentation provides new insight into the de novo germline mutations in mice, which merit publication. However, there remain several issues which the authors have to treat with.

Thankyou for your comments.

Issues:

Line 149-151 and Line 154-157:

Revise this correctly. The same sentence is repeated twice.

This comment refers to the section :

“The Variant Allele Fraction (VAF) for the observed VEE mutations in mice and humans were consistent with the vast majority occurring in the first cell division that contributes to the embryo and were highly concordant between different tissues. The number of VEE mutations per offspring varied considerably more than expected under a Poisson distribution ($p=0.0019$), suggesting this stage is more mutagenic for some zygotes than others. The Variant Allele Fraction (VAF) for the observed VEE mutations in mice and humans were consistent with the vast majority occurring in the first cleavage cell division that contributes to the embryo and were highly concordant between the two sequenced tissues.”

This has been changed to :

“The Variant Allele Fraction (VAF) for the observed VEE mutations in mice and humans were consistent with the vast majority occurring in the first cell division that contributes to the embryo and were highly concordant between different tissues. The number of VEE mutations per offspring varied considerably more than expected under a Poisson distribution ($p=0.0019$), suggesting this stage is more mutagenic for some zygotes than others.”

Line 263:

Remove “)”

This has been removed.

Line 267:

“We observed no difference between ~”

I think that Ex-data Fig6 shows potential difference in the spectra, albeit statistically no difference. So, the author should clarify the sentence. For example, “we found no statistically difference at least within our detection limit”.

This refers to the line “We observed 267 no difference between maternal and paternal mutation spectra in mice and humans 268 (Extended Data Figure 6).” We have amended this to

“We observed no statistically significant difference between maternal and paternal mutation spectra in mice and humans (Extended Data Figure 6).

“

Line 284-288:
Please clarify this sentence.

This refers to the following sentence:

“The rate of increasing mutations with parental age observed in mice is approximately five-fold greater ($p= 0.0003$) than in humans, which is larger than the two-fold more rapid rate of turnover of SSCs in mice compared to humans, implying a higher mutation rate per SSC division in mice compared to humans¹⁴”

This has been simplified to :

“Humans have a two-fold higher turnover in SSCs than mice, and given that the rate mutations accumulate in mice due to parental age is 5-fold higher than in humans, this implies that humans have a higher mutation rate per SSC division than mice^o”.

Line 307 and other place:

It is doubtful that mutation rate per cell division is the highest in the first cell division. The authors should remark the possibility that many of early embryonic cells would not contribute epiblast. If most of early embryonic cells lead to trophoblast or cell death, the discussion would be completely different. The authors should revise the manuscript to clarify this point.

This refers to the following sentence:

“Mutation rates per cell division are highest in the first cell division of embryonic development in both species. “

We agree with the reviewer that the first zygotic cell divisions may not contribute to the embryo, but extra-embryonic tissues. We have tried to make this clear in the text.

However, we have changed the sentence above to :

“Mutation rates per cell division are highest in the first cell division that contributes to the developing embryo in both species.”

Elsewhere we say “One notable similarity between mouse and human germlines was the hypermutability of the first post-zygotic cell division “*contributing to the developing embryo.*” We think this observation is likely to be real and has been replicated elsewhere (see Harland, C et al, Frequency of mosaicism points towards mutation-prone early cleavage cell divisions. Preprint at <https://doi.org/10.1101/079863> [biorxiv.org] (2017))

Line 458 and Line 535:

The same paper is listed twice as a reference.

This applies to Milholland, B et al. Differences between germline and somatic mutation rates in humans and mice. Nature communications 8:15183 | (2017). This has been fixed.

Line 794-:

I cannot understand the following formula to estimate per generation mutation rate.

$$u = m / p_{\text{depth}} p_{\text{filter}} p_{\text{dnm}}$$

I think that the mutation rate cannot be obtained by dividing the average number of mutations by percentages. Please clarify this. In addition, the authors should show all values (including 3 Ps) which were used in this study. These values must be helpful for readers to understand the calculation process and the accuracy of this analysis.

This applies to the following section:

We estimated the number of autosomal DNMs in each mouse offspring by correcting for the proportion of the genome that was interrogated as follows. Bedtools was used to calculate the proportion of the genome considered in our analysis after removing sites with low- or high-sequence depths for each individual (p_{depth}). We then calculated the proportion of sites that were retained after applying our whole-genome filters (simple sequence repeats and segmental duplications) after the depth filters were applied (p_{filter}). Last, we used the posterior probability supplied by DeNovoGear to calculate what proportion of true DNMs arose at sites that could be validated (p_{dnm}). Multisite variants were considered to be a single mutational event. The mutation rate was estimated as follows where \bar{m} is the average number of mutations observed per offspring.

$$\hat{\mu}_{\text{corrected}} = \bar{m} p_{\text{depth}} p_{\text{filter}} p_{\text{dnm}}$$

This section has been corrected to :

We estimated the number of autosomal DNMs in each mouse offspring by correcting for the proportion of the genome that was interrogated as follows. Bedtools³³ was used to calculate the proportion of the genome considered in our analysis after removing sites with low- or high-sequence depths for each individual (p_{depth}). We then calculated the proportion of sites that were retained after applying our whole-genome filters (simple sequence repeats and segmental duplications) after the depth filters were applied (p_{filter}). Last, we used the posterior probability supplied by *DeNovoGear* to calculate what proportion of true DNMs arose at sites that could be validated (p_{dnm}). Multisite variants were considered to be a single mutational event. The mutation rate was estimated as follows where m is the average number of mutations observed per offspring.

$$\text{corrected} = 100 * m / p\text{depthfilterp}dnm$$

The percentage of the genome covered after (pdepthfilterp) ranged from 85.7% to 91.9% with an average of 89.9%.

Reviewer #3 (Remarks to the Author):

The study design is novel and provides insight into where and when mutations occur between subsequent generations. The authors have improved presentation of their results and highlighted the difference between VEE mutations and the transmitted ones. However, I find that the authors have partially addressed my concerns regarding the presentation of the VEE mutations in the text. I realize, this is a non-trivial task to introduce these concepts to the general reader, however, here are some suggestions that could help the general reader to understand the distinction between the VEE mutations and the other classes, i.e. they most likely occurred in the offspring but not the parent.

In the temporal strata column in Fig 2A, be explicit in the text that EE are constitutional in offspring, mosaic in the parents, and so on for the other classes. Maybe add a dashed line to 2A, separating the VEE mutations from the other mutations. In figure 2B, the relationship of inner/outer circle to parent/child should be annotated.

We have added the following lines to the “temporal strata” box in Figure 2.

EE : “detectable as mosaic in parental tissues, constitutive in offspring.”

Peri-PGC “not detectable as mosaic in parental tissues, constitutive in offspring”

Late-post-PGC “absent in parents, constitutive in a single offspring”

VEE “ absent in parents, mosaic, not constitutive in offspring”

The authors insists incorporating the VEE mutations into the mutation rate analysis, they claim their imputation models their transmission rate reasonably well. However, given the non-transmitted nature of the VEE mutations I would want to see an extra column in Table 1 with the mutation rates calculated without the VEE mutations.

While we have not directly observed transmission of the VEE mutations we discovered in the offspring, we show in a separate experiment that VEE mutations observed in parents were transmitted to offspring. Therefore we believe mutation rate calculations

including the imputed VEE mutation rates are likely to be more accurate. On this basis we would prefer not to add this extra section unless the editor decides it is necessary.

Specific Comments

Lines 149-157: The VEE mutations are discovered by comparing them to binomial with $p=0.25$ and they were required to be concordant across tissues according to the methods (Lines 704-709). Therefore, I find that they are describing their criteria of detecting VEE mutations rather than novel results.

This comment applies to the following section:

“The Variant Allele Fraction (VAF) for the observed VEE mutations in mice and humans were consistent with the vast majority occurring in the first cell division that contributes to the embryo and were highly concordant between different tissues. The number of VEE mutations per offspring varied considerably more than expected under a Poisson distribution ($p=0.0019$), suggesting this stage is more mutagenic for some zygotes than others. The Variant Allele Fraction (VAF) for the observed VEE mutations in mice and humans were consistent with the vast majority occurring in the first cleavage cell division that contributes to the embryo and were highly concordant between the two sequenced tissues.”

This comment is not correct with regard to the discovery of VEE mutations in the offspring. They discovered by a de novo mutation caller with the expectation that the de novo mutation calls would be constitutive, and therefore the expectation is actually that they should be in 50% of cells, not 25%. The capturing of calls that deviate from 50% of cells will depend on factors such as trio sequencing depth but not an expectation that cells will contain 25% of the mutant allele. We did not require a call to be consistent between tissues to be made; the consistency of the VAF between the tissues is an observation of the underlying data not a result of filtering. The discovery of VEE mutations in parents was carried out using a different method described in the Methods section “Estimation of VEE contribution to germline mutation rate”, however, the only restriction we applied was that the VAF in the candidate tissue should be less than 35%.

Figure 6. I understand that the authors are reconstructing the parental germ cell lineages (diploid) through the haploid representation of the offspring. Furthermore, I understand that if the offspring inherits the non-mutated allele at the mutated loci, the offspring is not necessarily a carrier of mutation defining the diploid cell lineage. The authors define germline mosaic frequency (GMF) like this “Germline mosaic frequency was calculated as number of offspring carrying mutation/total number of offspring assayed” in the caption of Extended Data Table 2. Then a mutation present in all germ cells of the parent should have an expected GMF value of 50%. In light of this, it is improbable to have GMFs of 31.7% (P1; 62.4% of germ cells), 22% (P2; 44% of germ cells), 29.3% (P3; 58.6% of germ cells) and 17.1% (P4; 34.2% of germ cells) for mutually exclusive cell lineages from GPCB2 (father), as the sum is greater than 100% (200% of germ cells). These values indicate that these cell lineages are not mutually exclusive and therefore the topology of the cell lineages reported in Figure 6 should be revised. Is there perhaps a discrepancy how the GMF is calculated and how it is defined? For example for chr1: 195105306:C>T, the number of carriers is 8 out of 77 genotyped (GMF of 10.3%) but the reported GMF in Extended Data

Table 2 is 19.5%.

Thankyou for your persistence with this issue, our apologies for missing a discrepancy in the reported data. As you suggested, there was an error within the Extended Table 2 where the wrong cell was used as a reference cell for the calculation of GMF. The error resulted in inaccurate reporting of GMF in the last column of Extended Table 2. As the reviewer suggests, the GMF for chr1: 195105306:C>T should be 10.3% (8/77 offspring, reported in Extended Table 2 Number of offspring with mutation| Number of offspring sampled) and not 19.5% (reported in Extended Table 2 as GMF). With the correct values, the GMFs for the lineages in GPCB2 (father) is P6 11.7% (22% of cells), P7 16.9%, (33% of cells), P8 13.6% (~28% of cells), P9 8%, (~16% of cells) leading to a total of 100% of cells. Extended Table 2 (now Supplementary Table 2) has been corrected.

Minor comments

Line 52: Is this the correct reference (14)? I would guess the correct reference is Rahbari et al 2016. This work should also be referenced “Multiple transmissions of de novo mutations in families” Nature Genetics volume 50, 1674–1680 (2018).

Thankyou this is corrected.

Line 66: extra space after the citation.

This is corrected.

Line 80: There is no reference to Figure 2c in the text, maybe it could be referenced here

Thankyou for the suggestion, this has been added

Figure 1 caption: between 41 and 77 offspring, should be probably be 14 and 77 offspring.

41 is correct, unfortunately due to technical limitations we were only able to sequence a partial pedigree.

Lines 136-144: perhaps incorporate this into the introduction?

This refers to the section “Unlike in humans, mouse PGC specification is well characterised; after specification, PGCs proliferate rapidly, generating thousands of germ cell progenitors in both sexes . In the absence of strong positive selection, only mutations that occur prior to this proliferation are likely to be observed in multiple siblings in our pedigrees. In support of this assumption, studies of phenotypic markers of mutation have indicated that spermatogonial stem cells need to be depleted almost to extinction to result in sharing of phenotypes induced by later mutations among offspring.”

We think it may be better to include this section directly in the section where we are discussing timing of mutations in mice where it is directly relevant, but we are happy to move this section to the introduction if the editor advises.

Figure 4: the images are in low resolution.

This has now been fixed.

Figure 4: it would be nice to have the number of families and offspring in each panel.

We have put this information into the legend.

Line 396: the reference is not in the correct format.

Sorry, this has been fixed.

Reviewer #4 (Remarks to the Author):

I find the manuscript to be a marginal improvement (reads slightly better with some additional analyses) but it seems we have reached an impasse about my major concern from last time i.e. the comparison with human mutational data.

I still think that this is a significant study on germline mutation rates in mice, but the weak and underpowered comparisons with human data do not merit the current title or the conclusions highlighted in the abstract. This point has been raised by the other reviewer too, who clearly sees the scientific merit in this manuscript, but gets distracted by the novel vs expected conclusions.

I agree with the authors suggestion that the editor should make the final call about this.